# ROTATIONAL EQUILIBRIUM: HOW WEIGHT DECAY BALANCES LEARNING ACROSS NEURAL NETWORKS

## ABSTRACT

Weight decay can significantly impact the optimization dynamics of deep neural networks. In certain situations the effects of weight decay and gradient updates on the magnitude of a parameter vector cancel out on average, forming a state known as equilibrium. This causes the expected rotation of the vector in each update to remain constant along with its magnitude. Importantly, equilibrium can arise independently for the weight vectors of different layers and neurons. These equilibria are highly homogeneous for some optimizer and normalization configurations, effectively balancing the average rotation—a proxy for the effective learning rate—across network components. In this work we explore the equilibrium states of multiple optimizers including AdamW and SGD with momentum, providing insights into interactions between the learning rate, weight decay, initialization, normalization and learning rate schedule. We show how rotational equilibrium can be enforced throughout training, eliminating the chaotic transient phase corresponding to the transition towards equilibrium, thus simplifying the training dynamics. Finally, we show that rotational behavior may play a key role in the effectiveness of AdamW compared to Adam with $L_2$-regularization, the performance of different normalization layers, and the need for learning rate warmup.

## 1 INTRODUCTION

Modern neural networks are typically deep and structurally diverse compared to their predecessors, containing a variety of layer types and operations. The training of such networks requires simultaneously updating parameters used in many different contexts which poses additional challenges for optimization. For example, the gradients and activations must be kept in check to avoid effects such as vanishing gradients, and exploding activations. Efficient training intuitively also requires the learning of different components to be roughly balanced in some sense. A layer that is updated slowly, perhaps barely changing through the training process, is unlikely to contribute optimally to the final model resulting in worse performance and wasted compute. Conversely, a rapidly changing layer may cause instability, limiting the maximum stable learning rate and preventing other layers from learning effectively.

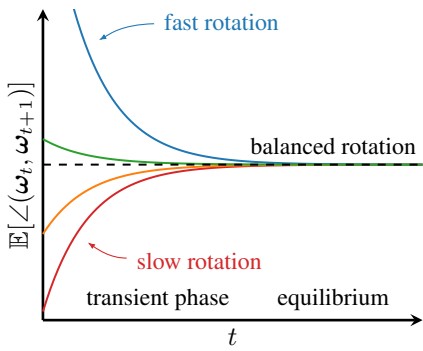

Figure 1: Conceptual figure of the angular updates of the weight vector $\omega_t$ for different normalized neurons (each line color) over time $t$ with a constant learning rate.

This suggests that sufficiently imbalanced rates are not optimal. Note however that there is nothing that says that perfectly balanced rates are optimal either, similar to how the use of a single learning rate across all layers may not be ideal in every case.

In this work we explore how weight decay can balance the updates for the weight vectors (but not biases) of different layers and neurons, measured in terms of the average relative update or angular change in each optimization step. This is caused by Spherical Motion Dynamics (Wan et al., 2021) which arise from the interaction of stochastic gradient updates and weight decay, especially in the presence of normalization layers like Batch Normalization (Ioffe and Szegedy, 2015). We describe the underlying mechanisms in more detail in Sections 2 and 3. Figure 1 shows how the average angular update size $\mathbb{E}[\angle(\omega_t, \omega_{t+1})]$ of different weight vectors $\omega$ could behave over time for typical

Spherical Motion Dynamics. Initially the rotation is somewhat arbitrary and is affected by the initial weight magnitude and potentially the magnitude of the gradient, depending on the optimizer used. Over time the weight norm converges to a stable equilibrium value in expectation, although the exact value can fluctuate between iterations. This causes the average angular update to have a specific magnitude, a state we call *rotational equilibrium*. For some setups the rotational equilibrium is identical for different layers and neurons, resulting in *balanced rotation* as depicted in Figure 1. Note that the formation of equilibrium does not require the convergence of the loss and occurs even for a neural network undergoing a random walk, which serves as the basis for our analysis.

This study expands upon prior investigations into the interactions between weight decay and normalization such as Van Laarhoven (2017); Zhang et al. (2019); Chiley et al. (2019); Li and Arora (2020), and particularly the Spherical Motion Dynamics (Wan et al., 2021). While we touch upon certain previous works throughout, please refer to Appendix A for an extended discussion. Earlier research has primarily focused on the general mechanisms and properties of weight decay and normalization, especially for plain Stochastic Gradient Descent, sometimes with heavy-ball momentum (Polyak, 1964). We focus on two new directions, rotational equilibrium in other optimizers like AdamW (Loshchilov and Hutter, 2019), and the importance of balanced rotation in the optimization of deep neural networks. Our main contributions can be summarized as:

- Presenting a simple and intuitive way to derive approximate expressions for the rotational equilibrium of the AdamW, Lion and SGD with momentum optimizers.
- Exploring how the weight decay and learning rate interact to form an effective step size schedule that differs from the standard learning rate schedule.
- Showing that AdamW results in balanced rotation unlike Adam with $L_2$ regularization and that this could explain the improved performance of AdamW observed in practice.
- Demonstrating that imbalanced rotation can degrade performance in more settings.
- Constructing rotational variants of existing optimizers that enforce balanced rotation, eliminating the transient phases and explicitly controlling the effective step size schedule, while also relating standard optimizers to relative optimizers like LARS.

## 2 PRELIMINARIES

### 2.1 NORMALIZATION AND SCALE-INVARIANCE

We say that a weight vector $\boldsymbol{\omega}$ is *scale-invariant* with respect to the loss $\mathscr{L}(\boldsymbol{\omega}, \ldots)$, which generally depends on other network parameters too, if scaling it by a positive factor does not affect the loss, i.e. $\mathscr{L}(r\boldsymbol{\omega}, \ldots) = \mathscr{L}(\boldsymbol{\omega}, \ldots), \forall r > 0$. When placed correctly relative to $\boldsymbol{\omega}$, many normalization operations such as Batch Normalization (Ioffe and Szegedy, 2015), Layer Normalization (Ba et al., 2016) and more (Huang and Belongie, 2017; Huang et al., 2017; Wu and He, 2018; Qiao et al., 2019) result in approximate scale-invariance for a given vector. Different forms of normalization can result in scale-invariance on different granularities, for example whole-layer for Layer Normalization and per-neuron for Batch Normalization when performed immediately following a given layer. Note that the concatenation of scale-invariant vectors is also scale-invariant. The gradient $\nabla_{\boldsymbol{\omega}}\mathscr{L}(\boldsymbol{\omega}, \ldots)$ for a scale-invariant vector $\boldsymbol{\omega}$ has two important properties:

$$\text{Gradient orthogonality:} \quad \nabla_{\boldsymbol{\omega}}\mathscr{L}(\boldsymbol{\omega}, \ldots) \perp \boldsymbol{\omega} \tag{1}$$

$$\text{Inverse proportionality:} \quad \|\nabla_{\boldsymbol{\omega}}\mathscr{L}(\boldsymbol{\omega}, \ldots)\| \propto \|\boldsymbol{\omega}\|^{-1} \tag{2}$$

When the weights are centered directly like in Weight Standardization (Qiao et al., 2019), we also have $\nabla_{\boldsymbol{\omega}}\mathscr{L}(\boldsymbol{\omega}, \ldots) \perp \mathbf{1}$. See Appendix B for further discussion about normalization and a derivation of these properties, which have also been described before (Li and Arora, 2020; Wan et al., 2021).

### 2.2 DEFINING MEASURES OF THE EFFECTIVE UPDATE SIZE

We use $\boldsymbol{\omega}$ to denote a weight vector that can achieve rotational equilibrium and refer to such vectors as *rotational weights*. They are typically used in dot products e.g. the weights of fully connected and convolutional layers and are often scale-invariant (but not always, see later sections). We use $\boldsymbol{p}$ for an arbitrary parameter that is not necessarily rotational. When training with weight decay, we break an update $\boldsymbol{p}_t \rightarrow \boldsymbol{p}_{t+1}$ (and analogously $\boldsymbol{\omega}_t \rightarrow \boldsymbol{\omega}_{t+1}$) for each parameter into two components:

$$\boldsymbol{p}_{t+1} - \boldsymbol{p}_t = \Delta_g\boldsymbol{p}_t + \Delta_\lambda\boldsymbol{p}_t \tag{3}$$

where $\Delta_{\boldsymbol{g}}\boldsymbol{p}_t$ comes from the loss gradient, which we denote with $\boldsymbol{g} := \nabla_{\boldsymbol{p}}\mathscr{L}(\boldsymbol{p}, \ldots)$, and $\Delta_\lambda\boldsymbol{p}_t$ from the weight decay or $L_2$ regularization. Both of these terms can include averaging over time like in SGD with momentum, causing them to depend on previous parameter and gradient values.

When measuring the size of an update in this work, we focus on the gradient component $\Delta_{\boldsymbol{g}}\boldsymbol{p}_t$, considering the weight decay as a separate process. Note that some scale-sensitive parameters like biases and gains are often excluded from weight decay, see for example the PyTorch Image Models (Wightman, 2019) or Hugging Face Transformers (Wolf et al., 2019) libraries. We can measure the update size of parameters using $\|\Delta_g\boldsymbol{p}_t\|$ or on average with the *root-mean-square (RMS) update size*:

$$\eta_g := \sqrt{\mathbb{E}[\|\Delta_g\boldsymbol{p}\|^2]} \tag{4}$$

For rotational weight vectors, we consider the *expected angular update size* defined as:

$$\eta_r := \mathbb{E}[\angle(\boldsymbol{\omega}_t, \boldsymbol{\omega}_{t+1})] = \mathbb{E}\left[\arccos\left(\frac{\langle\boldsymbol{\omega}_t, \boldsymbol{\omega}_{t+1}\rangle}{\|\boldsymbol{\omega}_t\|\|\boldsymbol{\omega}_{t+1}\|}\right)\right] \tag{5}$$

where $\langle\cdot\rangle$ denotes an inner product. Note that for a scale-invariant $\boldsymbol{\omega}$, only the direction $\boldsymbol{\omega}/\|\boldsymbol{\omega}\|$ matters since the magnitude $\|\boldsymbol{\omega}\|$ does not. As a result, $\eta_r$ is a more natural way to measure the effect of an update on $\boldsymbol{\omega}$, compared to other metrics such as $\eta_g$ that are not scale-invariant. We can also often scale one layer by a constant and undo the effects of this by scaling another parameter or layer (see e.g. Neyshabur et al. (2015)). In such cases relative metrics like the angular update may still capture the effect of an update better than $\eta_g$, even if $\boldsymbol{\omega}$ is not strictly scale-invariant by itself.

In related literature there are multiple definitions of an "effective" learning rate that vary slightly between works (Van Laarhoven, 2017; Chiley et al., 2019; Wan et al., 2021). We use $\eta_r$ as a measure of an *effective update size* that is closely related to these quantities. However, we want to emphasize the difference between measuring the size of a single update compared to the change over a longer time interval. The total weight change over extended periods is affected by both the magnitude of individual steps (i.e. $\eta_g$ or $\eta_r$) as well as the consistency in the update direction over the period which is affected by momentum and not captured by these metrics. The longer term change may be a better measure of an "effective" learning rate, but the step-wise metrics are easier to measure and control. For a given momentum coefficient they may be roughly proportional (e.g. in a random walk).

## 3 ANALYSIS

In this section we analyze the rotational equilibrium of a weight vector $\boldsymbol{\omega}$. The main goal is to obtain simple expressions for the equilibrium magnitude $\widehat{\|\boldsymbol{\omega}\|}$ and the expected angular update in equilibrium, $\widehat{\eta_r}$. To do this we analyze a simplified setting. Specifically, we assume the loss is in the form of empirical risk, i.e. $\mathscr{L}(\boldsymbol{\omega}, \mathbb{X}) = \frac{1}{|\mathbb{X}|}\sum_{\boldsymbol{x}\in\mathbb{X}}\mathscr{L}(\boldsymbol{\omega}, \boldsymbol{x})$ where $\mathbb{X}$ is our training dataset, $\boldsymbol{\omega}$ are our weights and $\boldsymbol{x}$ is a data point. The true noiseless gradient is then $\boldsymbol{g}_{\mathbb{X}} = \nabla_{\boldsymbol{\omega}}\mathscr{L}(\boldsymbol{\omega}, \mathbb{X})$ and the gradient for a minibatch $\mathbb{B}$ is $\boldsymbol{g}_{\mathbb{B}} = \nabla_{\boldsymbol{\omega}}\mathscr{L}(\boldsymbol{\omega}, \mathbb{B})$. We can define the noise in the gradient as $\boldsymbol{g}_N = \boldsymbol{g}_{\mathbb{B}} - \boldsymbol{g}_{\mathbb{X}}$ with $\mathbb{E}_{\mathbb{B}}[\boldsymbol{g}_N] = 0$ because $\mathbb{E}_{\mathbb{B}}[\boldsymbol{g}_{\mathbb{B}}] = \boldsymbol{g}_{\mathbb{X}}$ for a randomly sampled $\mathbb{B}$. Our analysis focuses on the case when the batch gradient is dominated by the noise component, i.e. $\boldsymbol{g}_{\mathbb{B}} = \boldsymbol{g}_{\mathbb{X}} + \boldsymbol{g}_N \approx \boldsymbol{g}_N$, resulting in a *random walk* for the neural network parameters. Appendix G gives further information and explores differences between a random walk and real neural network optimization and how they affect the predictions. In our experiments we find that the final predictions hold well for a variety of networks despite being derived for this simplified setting.

### 3.1 GEOMETRIC MODEL FOR EQUILIBRIUM

In this section we present a simple geometric derivation of the equilibrium norm $\widehat{\|\boldsymbol{\omega}\|}$ for different optimizers inspired in part by the analysis in Online Normalization (Chiley et al., 2019). Equilibrium is an abstract state where the effects of the gradient component of the update $\Delta_g\boldsymbol{\omega}$ and the weight decay component $\Delta_\lambda\boldsymbol{\omega}$ on the expected weight magnitude balance out on average. These components typically have different monotonic dependencies on the weight magnitude, with weight decay being proportional while the gradient component is either constant or inversely proportional, depending on the setting. As a result, the effects of these components can balance out in expectation for a

Figure 2: Weight norm equilibrium where a single expected optimizer step is split into components $\Delta_g\boldsymbol{\omega}$ due to the loss gradient and $\Delta_\lambda\boldsymbol{\omega}$ from weight decay.

Table 1: Analytical predictions for different optimizers and a parameter $\boldsymbol{p} \in \mathbb{R}^C$ with a gradient $\boldsymbol{g}$ and $\tilde{\boldsymbol{g}} := \|\boldsymbol{p}\|\boldsymbol{g}$. The **RMS update size** $\widehat{\eta}_g$ applies to all parameters, the **expected angular update** $\eta_r$ and **equilibrium norm** $\widehat{\|\boldsymbol{\omega}\|}$ only apply to a scale-invariant $\boldsymbol{\omega}$ in equilibrium.

| | SGDM (42) | AdamW (9) | Adam+$L_2$ (75) | Lion (54) |
|---|---|---|---|---|
| $\widehat{\eta}_g$ | $\eta\sqrt{\frac{\mathbb{E}[\|\boldsymbol{g}\|^2]}{1-\alpha^2}}$ | $\eta\sqrt{C\frac{1-\beta_1}{1+\beta_1}}$ | $\eta\sqrt{C\frac{1-\beta_1}{1+\beta_1}}$ | $\eta\sqrt{C}$ |
| $\widehat{\eta}_r$ | $\sqrt{\frac{2\eta\lambda}{1+\alpha}}$ | $\sqrt{2\eta\lambda\frac{1-\beta_1}{1+\beta_1}}$ | $\sqrt[3]{\frac{2\eta^2\lambda}{\langle\mathbf{1},\sqrt{\mathbb{E}[\tilde{\boldsymbol{g}}^2]}\rangle}}\sqrt{\frac{1-\beta_1}{1+\beta_1}C}$ | $\sqrt{\pi\eta\lambda}\left((1-\beta_1)^2+\beta_1^2\frac{1-\beta_2}{1+\beta_2}\right)^{\frac{1}{2}}$ |
| $\widehat{\|\boldsymbol{\omega}\|}$ | $\sqrt[4]{\frac{\eta\mathbb{E}[\|\tilde{\boldsymbol{g}}\|^2]}{2\lambda\cdot(1-\alpha)}}$ | $\sqrt{\frac{\eta C}{2\lambda}}$ | $\sqrt[3]{\frac{\eta}{2\lambda}\cdot\langle\mathbf{1},\sqrt{\mathbb{E}[\tilde{\boldsymbol{g}}^2]}\rangle}$ | $\sqrt{\frac{\eta C}{\pi\lambda}}\left((1-\beta_1)^2+\beta_1^2\frac{1-\beta_2}{1+\beta_2}\right)^{-\frac{1}{2}}$ |

particular magnitude, which we call the equilibrium norm $\widehat{\|\boldsymbol{\omega}\|}$. As shown in Figure 2, the geometry of this is not necessarily simple. Due to the averaging effects of momentum over time, $\Delta_g\boldsymbol{\omega}$ is not necessarily orthogonal to the weights even in cases where individual gradients are (e.g. for scale-invariant weights). In the same way, the weight decay (or $L_2$-regularization) component $\Delta_\lambda\boldsymbol{\omega}$ may not be perfectly anti-parallel to the weights with momentum.

To simplify the effects of momentum, we instead consider a different view of equilibrium shown in Figure 3. Here we consider the total weight change throughout training derived from the weight and gradient at a given time step, instead of the update that is applied in that iteration. We thus define $\boldsymbol{u}$ for time step $t$ as the sum of the contributions of $\nabla_{\boldsymbol{\omega}}\mathscr{L}(\boldsymbol{\omega}_t,\ldots)$ to subsequent updates $\boldsymbol{\omega}_t \to \boldsymbol{\omega}_{t+1}$, $\boldsymbol{\omega}_{t+1} \to \boldsymbol{\omega}_{t+2}$, and so on. Analogously, the weight decay term $\boldsymbol{d}$ is defined as the total weight change due to the weight decay or $L_2$-regularization of the weights $\boldsymbol{\omega}_t$ at iteration $t$. Note that without momentum $\boldsymbol{u} = \Delta_g\boldsymbol{\omega}$, $\boldsymbol{d} = \Delta_\lambda\boldsymbol{\omega}$ and that if $\Delta_g\boldsymbol{\omega}$ and $\Delta_\lambda\boldsymbol{\omega}$ balance out on average, then so must $\boldsymbol{u}$ and $\boldsymbol{d}$.

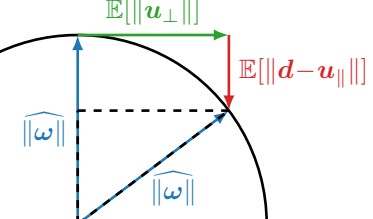

Figure 3: Weight norm equilibrium. The loss gradient causes an update $\boldsymbol{u}$ and the weight decay $\boldsymbol{d}$.

In many cases $\boldsymbol{u}$ is orthogonal to the weights on average due to scale-invariance or randomness, but otherwise can we split it into orthogonal $\boldsymbol{u}_\perp$ and radial $\boldsymbol{u}_\|$ components. The weight decay term $\boldsymbol{d}$ is anti-parallel to the weights in every case we consider. If we can obtain an expression for $\|\boldsymbol{u}_\perp\|$ and $\|\boldsymbol{d} - \boldsymbol{u}_\||$, this allows us to apply the Pythagorean theorem to the dashed triangle in Figure 3:

$$(\widehat{\|\boldsymbol{\omega}\|} - \mathbb{E}[\|\boldsymbol{d}-\boldsymbol{u}_\||])^2 + \mathbb{E}[\|\boldsymbol{u}_\perp\|]^2 = \widehat{\|\boldsymbol{\omega}\|}^2 \tag{6}$$

We can then solve for $\widehat{\|\boldsymbol{\omega}\|}$ after accounting for the dependency of $\boldsymbol{u}$ and $\boldsymbol{d}$ on the weight norm.

Once we have an expression for $\widehat{\|\boldsymbol{\omega}\|}$, we can compute a prediction for the RMS update size $\widehat{\eta}_g$. Combining $\widehat{\eta}_g$ for the equilibrium magnitude $\widehat{\|\boldsymbol{\omega}\|}$ allows us to compute the expected relative update size $\tilde{\eta}_g/\widehat{\|\boldsymbol{\omega}\|}$ which closely approximates $\eta_r$ in equilibrium. We do this for AdamW in the next subsection and for SGDM and Lion (Chen et al., 2023) in Appendix C and D. The results for each optimizer are summarized in Table 1.

### 3.2 ADAMW EQUILIBRIUM

The standard version of AdamW (Loshchilov and Hutter, 2019) can be written as:

$$\boldsymbol{m}_t = \beta_1\boldsymbol{m}_{t-1} + (1-\beta_1)\boldsymbol{g}_t \tag{7}$$

$$\boldsymbol{v}_t = \beta_2\boldsymbol{v}_{t-1} + (1-\beta_2)\boldsymbol{g}_t^2 \tag{8}$$

$$\boldsymbol{p}_t = \boldsymbol{p}_{t-1} - \eta \cdot \left(\frac{\boldsymbol{m}_t/(1-\beta_1^t)}{\sqrt{\boldsymbol{v}_t/(1-\beta_2^t)}+\varepsilon} + \lambda\boldsymbol{p}_{t-1}\right) \tag{9}$$

Where $\boldsymbol{p}_t \in \mathbb{R}^C$ is a parameter vector at time $t$, $\boldsymbol{g}_t = \nabla_{\boldsymbol{p}}\mathscr{L}(\boldsymbol{p}_t,\ldots)$ is the gradient, $\boldsymbol{m}$ is the first moment and $\boldsymbol{v}$ is the second moment. The learning rate ($\eta \geq 0$), weight decay ($\lambda \geq 0$), moment coefficients ($0 < \beta_1 < 1, 0 < \beta_2 < 1$) and $\varepsilon \geq 0$ are hyperparameters. For simplicity we assume that $\varepsilon$ and the bias correction can be ignored, i.e. that $\varepsilon$, $\beta_1^t$ and $\beta_2^t$ are all effectively zero.

**Equilibrium magnitude**: For a rotational weight $\boldsymbol{\omega}$, we can write $\boldsymbol{u}$ and $\boldsymbol{d}$ from Section 3.1 as:

$$\boldsymbol{u} = -\eta \sum_{k=t}^{\infty} \beta_1^{k-t}(1-\beta_1)\frac{\boldsymbol{g}_t}{\sqrt{\boldsymbol{v}_k}}, \qquad \boldsymbol{d} = -\eta\lambda\boldsymbol{\omega} \tag{10}$$

We note that due to symmetry, each coordinate of $\boldsymbol{u}$ has a zero-mean distribution in the random walk setup. Since $\boldsymbol{u}$ is independent from $\boldsymbol{\omega}$, this makes them orthogonal in expectation i.e. $\mathbb{E}[\langle \boldsymbol{u}, \boldsymbol{\omega} \rangle] = 0$. When the gradient distribution is not changing over time, it is also reasonable to assume that the variance of each coordinate remains constant resulting in $\forall t, k : \mathbb{E}[\|\boldsymbol{g}_t/\sqrt{\boldsymbol{v}_k}\|^2] = C$ (for $\boldsymbol{\omega} \in \mathbb{R}^C$) and by extension $\mathbb{E}[\|\boldsymbol{u}\|^2] = \eta^2 C$. Defining $\omega = \|\boldsymbol{\omega}\|$, $u = \|\boldsymbol{u}\|$, $u_\| = \langle \boldsymbol{\omega}, \boldsymbol{u} \rangle / \|\boldsymbol{\omega}\|$, $u_\perp^2 = u^2 - u_\|^2$ and $d = \|\boldsymbol{d}\|$ we can write a recurrence relation based on Equation (6):

$$\mathbb{E}[\omega_{i+1}^2] = \mathbb{E}[(\omega_i - d + u_\|)^2 + u_\perp^2] \tag{11}$$

$$= \mathbb{E}[\omega_i^2 - 2d\omega_i + 2u_\|\omega_i - 2du_\| + d^2 + u_\|^2 + (u^2 - u_\|^2)] \tag{12}$$

$$= \mathbb{E}[\omega_i^2](1 - 2\eta\lambda + \eta^2\lambda^2) + \eta^2 C \tag{13}$$

where we have used independence, $\mathbb{E}[u_\|] = 0$ and $\mathbb{E}[u] = \eta^2 C$. The solution is:

$$\mathbb{E}[\omega_i^2] = \mathbb{E}[\omega_0^2]a^i + \frac{\eta^2 C}{2\eta\lambda - \eta^2\lambda^2}(1 - a^i), \qquad a = (1 - 2\eta\lambda + \eta^2\lambda^2) \tag{14}$$

The recurrence relation is written in terms of $\boldsymbol{u}$ and $\boldsymbol{d}$ instead of $\Delta_g\boldsymbol{\omega}$ and $\Delta_\lambda\boldsymbol{\omega}$. This is thus only an approximation of how the real system converges to equilibrium over time, but still informative. It may be a good approximation if $\|\boldsymbol{\omega}\|$ changes slowly compared to how quickly $\boldsymbol{u}$ is applied (i.e. $\boldsymbol{m}$ changes) and $\boldsymbol{v}$ is updated. In either case, the limit gives us the equilibrium norm listed in Table 1:

$$\widehat{\|\boldsymbol{\omega}\|} = \sqrt{\frac{\eta C}{2\lambda - \eta\lambda^2}} \approx \sqrt{\frac{\eta C}{2\lambda}} \qquad \text{(for $\lambda\eta \ll 2$)} \tag{15}$$

**Update size:** We can estimate the RMS update size $\eta_g$ of $\Delta_g\boldsymbol{p} = \frac{\eta\boldsymbol{m}_t}{\sqrt{\boldsymbol{v}_t}}$ as follows:

$$\mathbb{E}[\|\Delta_g\boldsymbol{p}\|^2] = \mathbb{E}[\|\frac{\eta}{\sqrt{\boldsymbol{v}_t}}(1 - \beta_1)\sum_{k=0}^{t-1}\beta_1^{t-k}\boldsymbol{g}_{t-k}\|^2] \tag{16}$$

$$= \eta^2(1 - \beta_1)^2\sum_{k=0}^{t-1}\beta_1^{2t-2k}\mathbb{E}[\|\frac{\boldsymbol{g}_{t-k}}{\sqrt{\boldsymbol{v}_t}}\|^2] \tag{17}$$

$$\approx \eta^2\frac{1-\beta_1}{1+\beta_1}C \tag{18}$$

where we have approximated the geometric sum with its limit $t \to \infty$, used the fact that for the random walk $\forall j \neq k : \mathbb{E}[\langle \boldsymbol{g}_j, \boldsymbol{g}_k \rangle] = 0$ as well as our previous assumption $\forall t, k : \mathbb{E}[\|\boldsymbol{g}_t/\sqrt{\boldsymbol{v}_k}\|^2] = C$. This gives us the prediction $\widehat{\eta}_g \approx \mathbb{E}[\sqrt{\|\Delta_g\boldsymbol{p}\|^2}]$ listed in Table 1. Approximating the equilibrium angular update size with the expected relative update size $\sqrt{\mathbb{E}[\|\Delta_g\boldsymbol{\omega}\|^2]}/\widehat{\|\boldsymbol{\omega}\|}$ gives the $\widehat{\eta}_r$ value. This approximation is good for small relative updates and a relatively small radial component in $\Delta_g\boldsymbol{\omega}$.

### 3.3 Decoupled Weight Decay vs $L_2$-Regularization in Adam

Loshchilov and Hutter (2019) proposed the use of decoupled weight decay instead of $L_2$-regularization in Adam (Kingma and Ba, 2015). In their experiments they find that Adam with decoupled weight decay (i.e. AdamW, see Equation 9) outperforms the $L_2$-regularized form (i.e. Adam+$L_2$, Equation 75) across a wide range of settings. Since then AdamW has been widely adopted, but as far as we know the reason for its effectiveness over Adam+$L_2$ is not well understood.

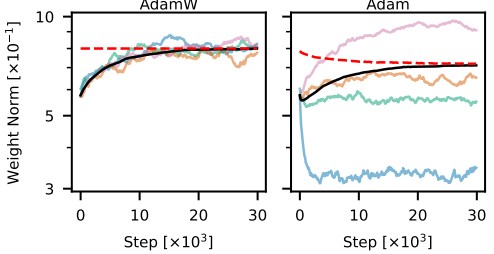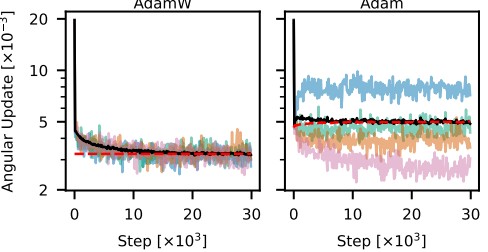

Figure 4: Comparing the equilibrium behavior of AdamW and Adam+$L_2$ in a simplified random setup for weight norm (left) and angular updates (right). Colors (pink, orange, green, blue) represent scale-invariant weight vectors of batch normalized neurons with varying gradient norms. Black lines represent layer-wide averages, and red dashed lines show their equilibrium predictions from Table 1.

In Appendix E we analyze the geometric model for Adam+$L_2$, revealing that the both the equilibrium norm and angular update size depend on the gradient magnitude, unlike AdamW (see Table 1). When the gradient norm varies between neurons or layers, this results in imbalanced rotation. Figure 4 shows an example of this for a random walk in a simple network described in Appendix F. We believe the balanced vs imbalanced equilibrium rotation is a key difference between Adam+$L_2$ and AdamW which may explain why decoupled weight decay is more effective for Adam-like methods. We explore this further in our experiments.

### 3.4 Rotational Dynamics of Scale-Sensitive Parameters

Prior work has primarily focused on the dynamics of scale-invariant weights. Note that any weight vector can be made scale-invariant by simply applying normalization to it, for example in the form of Weight Standardization (Qiao et al., 2019). For a random walk the gradient component is always orthogonal in expectation, but for real tasks scale-sensitive weights sometimes have an average radial gradient component $\mathbb{E}[\boldsymbol{u}_{\|}] \neq 0$. In Appendix H we explore how this affects the rotational dynamics of these weights (for SGDM). We find that a radial component acts like an additional weight decay $\lambda_u = -\mathbb{E}[\langle \boldsymbol{u}, \boldsymbol{\omega} \rangle]/(\eta \|\boldsymbol{\omega}\|^2)$ that can be combined with $\lambda$ to give a new "effective" weight decay $\lambda_e = \lambda + \lambda_u$, resulting in dynamics similar to scale-invariant weights with the adjusted value.

## 4 Rotational Variants of Optimizers (RVs)

In previous sections we have discussed rotational equilibrium in standard optimizers like AdamW. Training with these optimizers requires the weights to transition towards equilibrium at the start of training and when hyperparameters such as the learning rate change. This creates transient phases where rotation can be imbalanced (see e.g. Figure 1) and far away from the equilibrium value, causing a mismatch between the specified learning rate schedule and the resulting effective step sizes over time. Standard optimizers also rely on proper normalization to avoid radial gradient components that can cause imbalanced rotation in equilibrium, see Section 3.4. In this section we create an experimental tool to study these phenomena. Algorithm 1 shows how we can create rotational variants of existing optimizers by explicitly controlling the average angular update size, forcing it to match equilibrium throughout training. This eliminates the transient phases completely and can balance the rotation of different layers and neurons without relying on normalization. By default we target the equilibrium dynamics of Weight Standardization (Qiao et al., 2019). We keep the weight magnitude constant and optionally introduce a learnable gain to compensate, which can matter for scale-sensitive weights and avoids numerical issues. Further details about the rotational wrapper can be found in Appendix I. Note that the form of the rotational wrapper closely resembles that of relative optimizers like LARS (You et al., 2017) and others discussed in Appendix A, revealing a connection to the equilibrium dynamics of standard optimizers.

---

**Algorithm 1** Our proposed Rotational Wrapper for training with constrained rotational dynamics.

---

**Require:** Inner optimizer F, decay factor $0 \leq \beta < 1$, $\varepsilon \geq 0$ for numerical stability, iteration count $T$, set $\Omega$

1: **for** $\boldsymbol{p}$ in $\Omega$ **:**           ▷ *For weights we choose to treat as rotational*
2:     $\nu_{\boldsymbol{p}} \leftarrow 0$           ▷ *Initialize the update RMS tracker*
3:     $n_{\boldsymbol{p}} \leftarrow \|\boldsymbol{p}\|$           ▷ *Save the initial magnitude*
4: **for** $t \in \{1, ..., T\}$ **:**
5:     Perform backpropagation, obtain gradients for all parameters
6:     **for all** $\boldsymbol{p}$ **:**           ▷ *For each parameter*
7:        $\Delta_g \boldsymbol{p}, \Delta_\lambda \boldsymbol{p} \leftarrow \text{F.get\_update}(\boldsymbol{p}, \nabla_{\boldsymbol{p}} \mathscr{L}(\boldsymbol{p}, \ldots))$       ▷ *Get update components (Equation (3))*
8:        **if** $\boldsymbol{p} \in \Omega$ **:**           ▷ *If $\boldsymbol{p}$ should be treated as rotational*
9:           $\Delta_g \boldsymbol{p} \leftarrow \Delta_g \boldsymbol{p} / \eta$           ▷ *Remove the effect of the learning rate $\eta$ used in F*
10:           $\Delta_g \boldsymbol{p} \leftarrow \Delta_g \boldsymbol{p} - \frac{\langle \Delta_g \boldsymbol{p}, \boldsymbol{p} \rangle}{\|\boldsymbol{p}\|^2} \boldsymbol{p}$        ▷ *Remove the projection onto $\boldsymbol{p}$, making $\Delta_g \boldsymbol{p} \perp \boldsymbol{p}$*
11:           $\nu_{\boldsymbol{p}} \leftarrow \beta \cdot \nu_{\boldsymbol{p}} + (1 - \beta) \cdot \|\Delta_g \boldsymbol{p}\|^2$        ▷ *Update RMS tracker*
12:           $\boldsymbol{p} \leftarrow \boldsymbol{p} + \widehat{\eta_r} \cdot n_{\boldsymbol{p}} \cdot \frac{\Delta_g \boldsymbol{p}}{\sqrt{\nu_{\boldsymbol{p}}/(1 - \beta^t)} + \varepsilon}$     ▷ *Rotate $\boldsymbol{p}$ by $\widehat{\eta_r}$ from Table 1 on average*
13:           $\boldsymbol{p} \leftarrow n_{\boldsymbol{p}} \cdot \frac{\boldsymbol{p} - \bar{\boldsymbol{p}}}{\|\boldsymbol{p} - \bar{\boldsymbol{p}}\|}$        ▷ *Center and normalize $\boldsymbol{p}$ to the initial magnitude*
14:        **else:**           ▷ *Treat $\boldsymbol{p}$ as non-rotational*
15:           $\boldsymbol{p} \leftarrow \boldsymbol{p} + \Delta_g \boldsymbol{p} + \Delta_\lambda \boldsymbol{p}$           ▷ *Perform standard update*

---

Table 2: Test set performance (mean±std) over three seeds for the baseline optimizer, AdamW, and its rotational variant (RV). We use the baseline hyperparameters directly for the zero-shot results. For the best shot results, minor tuning was applied. The final column shows results for Adam+$L_2$ directional updates with a balanced angular update speed $\widehat{\eta}_r$ based on AdamW across all neurons. †The baseline $\lambda$ is too low, causing an extended transient phase where the RV rotation differs.

| Dataset | Model | Batch Size | Metric ($\updownarrow$) | AdamW Baseline | RV-AdamW Zero Shot | RV-AdamW Best Shot | Wrapped Adam+$L_2$ |
|---|---|---|---|---|---|---|---|
| CIFAR-10 | ResNet-20 | 128 | Top-1 Acc. ($\uparrow$) | 92.2 ±0.11 | 92.3±0.25 | N/A | 92.3±0.18 |
| CIFAR-10 | ResNet-20 | 2048 | Top-1 Acc. ($\uparrow$) | 91.5 ±0.22 | 91.2 ±0.21 | 91.9 ±0.29 | 91.8±0.15 |
| CIFAR-10 | DeiT tiny | 64 | Top-1 Acc. ($\uparrow$) | 95.9 ±0.07 | 96.3±0.25 | N/A | 96.3±0.17 |
| Imagenet-1k | DeiT tiny | 1024 | Top-1 Acc. ($\uparrow$) | 72.1 | 71.5 | 72.3 | N/A |
| IWSLT2014 de-en | Transformer-S | 4096 | Bleu ($\uparrow$) | 34.6±0.06 | 19.9±0.14† | 34.7±0.10 | 34.5±0.04 |
| Wikitext | GPT-like | 55 | Perplexity ($\downarrow$) | 19.6 ±0.07 | 19.1±0.21 | N/A | 19.3±0.12 |

## 5 EXPERIMENTS

**Experimental Setup:** We conducted experiments on several well-known datasets and standard network architectures (details in Appendix K). When using our RVs, we apply rotational updates to all convolutional and linear layers. For transformers, these layers are not always fully scale-invariant; the weight norms matter. To address this, we introduced learnable gains as mentioned in Section 4.

**Constraining the Rotational Dynamics:** We explore the impact of constraining the rotational dynamics by comparing the performance of AdamW and RV-AdamW across various network architectures and tasks, see Table 2. The RVs achieve comparable performance to the original versions without any additional hyperparameter tuning (zero-shot) or light tuning (best-shot). This suggests that the simplified learning dynamics (e.g. no transient phase) with the RVs are sufficient. Although the RVs have many interesting properties and good performance, we view them primarily as a research tool.

**Learning Rate vs Weight Decay:** Different combinations of a learning rate $\eta$ and weight decay $\lambda$ with a constant product $\eta\lambda$ result in the same expected angular update $\widehat{\eta}$ in equilibrium but different RMS update sizes $\widehat{\eta}_g$ (see Table 1). This affects the rotational weights (especially when scale-invariant) differently than gains and biases (which have $\lambda=0$). Figure 5 (left) explores the impact of this. For small $\eta$, biases and gains update slowly since $\widehat{\eta}_g \propto \eta$, resulting in accuracy comparable to freezing them (90.8%). Conversely, large $\eta$ results in large updates to these parameters, potentially causing unstable training. We attribute the performance difference between AdamW and the RV to varying effective update size schedules (see Figure 5 middle and right).

**Scheduling Effects:** In Figure 5 (right), there are noticeable deviations between the measurements of $\|\boldsymbol{\omega}\|$ and $\eta_r$ and the predictions listed in Table 1 for a cosine decay learning rate schedule and a five epoch warmup. In the initial phase we observe the transient phase, corresponding to the transition towards equilibrium, while the fall off in the end phase indicates that the learning rate is too small for the weights to decay fast enough to maintain equilibrium. The same effective step size schedule could be achieved by the RV with an adjusted learning rate schedule. See Appendix K for details.

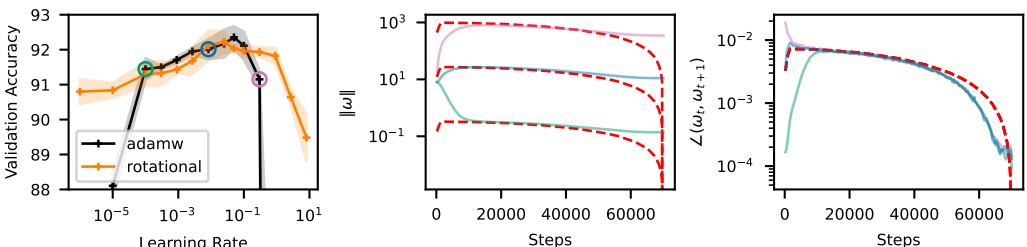

Figure 5: **Left:** Validation accuracy for ResNet-20 on CIFAR-10 for different learning rate, weight decay pairs with a constant product ($\eta\lambda = 5\cdot10^{-4}$) resulting in a specific $\widehat{\eta}_r$ (Table 1). **Right:** The weight norm $\|\boldsymbol{\omega}\|$ and angular update size $\eta_r$ over time for three $(\eta, \lambda)$ pairs corresponding to the colored circles on the left with predictions in dashed red. Note the difference in the initial/final phase.

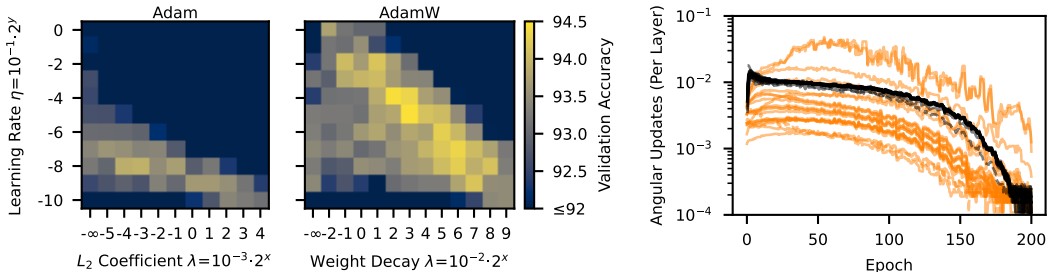

Figure 6: **Left:** A hyperparameter sweep for the learning rate and $L_2$ regularization / weight decay of Adam and AdamW on CIFAR-10 ResNet-18. Adam is unable to match the performance of AdamW. **Right:** The average angular per-step updates (radians) of each layer over the course of training for the best configuration of Adam (orange) and AdamW (black). In both cases the updates change with the learning rate schedule (cosine, no warmup) but for Adam they vary considerably across layers. The dashed line corresponds to the final fully-connected layer which is scale-sensitive.

**Adam vs AdamW:** Figure 6 (left) reproduces the performance gap between AdamW and Adam+$L_2$ observed by Loshchilov and Hutter (2019), around 0.5% on the validation set. The right panel shows that Adam+$L_2$ results in an unbalanced rotation unlike AdamW, confirming our observations from Section 3.3. To determine whether this contributes to the performance gap, we create a special RV that combines the update direction $\Delta_g \boldsymbol{p}$ from Adam+$L_2$ with the $\widehat{\eta_r}$ of AdamW, ensuring balanced angular updates across all neurons and layers. For the experiment in Figure 6, the special RV performs identically to a standard RV-AdamW, outperforming Adam+$L_2$ by roughly 0.5%. In the final column of Table 2 we show that this holds for more settings, with the special RV and RV-AdamW performing similar, roughly matching or outperforming AdamW in all cases.

**Imbalanced Rotation:** To further quantify the impact of imbalanced rotation we experiment with artificially scaling $\eta_r$ for a fraction of the neurons in each layer. Figure 7 shows the result of varying the fraction (middle) and scale for half the neurons (right), after tuning the learning rate for each configuration. We observe that even small variations in $\eta_r$ can significantly affect performance.

**Training Poorly Normalized Networks:** Layer Normalization can make a whole layer scale-invariant but not individual neurons (unlike e.g. Batch Normalization). For standard optimizers, this can result in imbalanced rotation across neurons but Algorithm 1 ensures balanced rotation irrespective of the normalization. Figure 8 (left) shows that this results in improved performance across learning rates when training a layer-normalized ResNet-18 on CIFAR-100. We also observed a slight performance increase for the small GPT-like model in Table 1, potentially for a similar reason.

**Need for Learning Rate Warmup:** For standard optimizers the initial transient phase can result in imbalanced and overly fast rotation that does not match the learning rate schedule. We conjecture that the common practice of learning rate warmup is often beneficial in part by counteracting this effect. In Figure 8 (right) we train a ResNet-50 on ImageNet for 10 epochs using large batch sizes and different learning rates without warmup. The RV is more stable and achieves higher accuracy.

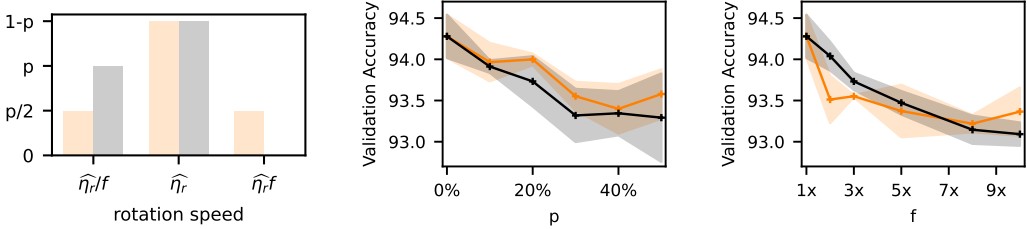

Figure 7: **Left:** Two artificially imbalanced angular update size distributions (black/orange). A portion $p$ of the neurons is rotating $f$ times slower and/or faster than rest using a modified RV for ResNet-18 training on CIFAR-10. **Middle:** Varying $p$ for a fixed $f = 10$. The performance with $p = 50\%$ is comparable to a network half the width (93.5%). **Right:** Varying $f \in [1, 10]$ for $p = 50\%$.

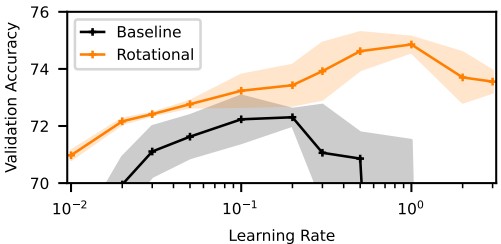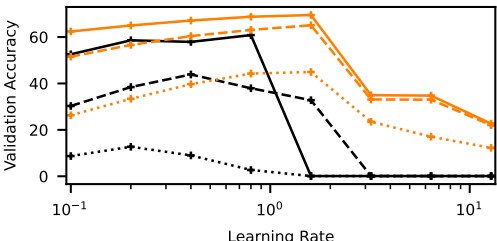

Figure 8: Comparison of the final validation accuracy of runs with different learning rates, using either SGDM (black) or RV-SGDM (orange). **Left:** Layer normalized ResNet-18 training on CIFAR-100, accuracy given as (mean±std of 5 runs). **Right:** 10 epoch training of ResNet-50 on ImageNet-1k without a learning rate warmup using large batch sizes (2k solid, 8k dashed, 32k dotted).

## 6 DISCUSSION & CONCLUSION

We believe rotational dynamics can provide valuable insights into many phenomena in deep learning optimization. Our analysis shows how the average angular update size $\eta_r$ in equilibrium and the RMS gradient size $\eta_g$ have a different dependency on the learning rate $\eta$ and weight decay $\lambda$ (Table 1). In the common setting where gains and biases are excluded from weight decay, this results in a scaling of their "effective" update size, $\eta_g$, compared to that of the rotational weights, $\eta_r$. In equilibrium, $\eta$ and $\lambda$ therefore jointly determine two effective update sizes used for different types of parameters. Curiously, $\eta_r \propto \sqrt{\eta}$ while $\eta_g \propto \eta$, raising questions about the impact of this when varying the learning rate $\eta$ with a schedule or scaling the batch size. We do not explore this here but note that this is not the case in most relative optimizers, such as Nero (Liu et al., 2021) for example.

In our analysis and experiments we found that the equilibrium in Adam with $L_2$ regularization significantly differs from that of AdamW and other optimizers, as the expected angular update in equilibrium depends on the gradient magnitude. This results in unbalanced rotation across vectors that have different gradient norms. Forcing Adam+$L_2$ to have balanced rotation through our rotational wrapper seems to eliminate most of the performance degradation compared to AdamW.

Normalization can make weights scale-invariant, ensuring the gradients are orthogonal to the weights. A consistent gradient component parallel to a weight vector results in a faster or slower rotation in equilibrium (Section 3.4), potentially resulting in unbalanced rotation across different vectors. Different types of normalization result in different granularity of scale-invariance e.g. layer level with Layer Normalization and neuron level with Batch Normalization and Weight Standardization. With more coarse normalization operations this can result in unbalanced equilibrium rotation on the neuron level. We observe that the use of Rotational Optimizer Variants can improve the performance of layer normalized ResNets and also the GPT-like model, likely by enforcing balanced rotation. This could also help explain how Weight Standardization helps aid optimization when applied on top of Layer Normalization or Group Normalization, i.e. by ensuring scale-invariance on a finer level.

In standard optimizers different weight vectors must converge to equilibrium. Depending on the initialization weight magnitude, learning rate, weight decay and gradient norm (for SGDM only), this can result in faster or slower angular updates in the initial transient phase. This causes a mismatch between the specified learning rate schedule and the resulting angular step sizes over time. A learning rate warmup may potentially be needed in part to counteract fast initial rotation due to this effect. Using the rotational optimizer variants (RVs) eliminates this effect, ensuring angular updates follow the specified learning rate schedule. This may also explain why other relative optimizers like LARS seem to have a reduced need for learning rate warmup. Although this may not be the sole reason a warmup is needed in every case, we believe it is an important effect to be aware of.

In this project we have explored the importance of rotational dynamics in deep learning optimization aiming for a high level understanding. We hope our insights can be of use to practitioners when debugging or tuning neural network training, and that they may provide a new perspective for theorists. We see a lot of potential for future work in this area in terms of more formal and rigorous theory as well as practical application to e.g. optimizer and scheduler design.

## 7 REPRODUCIBILITY

The code used for our experiments is available at `https://github.com/fZuDIKrLqR/rotational-equilibrium`. We provide more experimental details in Appendix K including hyperparameters not listed in the main body. The analysis for the other optimizers are also provided in the Appendix, SGDM in Appendix C, Lion in Appendix D and Adam+$L_2$ in Appendix E. Additional information about the analytical setting used in Section 3 and how it compares to practical settings can be found in Appendix G.

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

## A    EXPANDED RELATED WORK

In this section we discuss more related works divided into five main categories.

### A.1    UNDERSTANDING AND IMPROVING WEIGHT DECAY

We will add the main works in this area as well as promised references in a future revision.

### A.2    SCALE-INVARIANCE AND EFFECTIVE LEARNING RATES

Several works have investigated how the scale-invariance results in a certain "effective learning rate" based on the relative change that varies based on the norm of the weights, often in the form of $\eta/\|\boldsymbol{\omega}\|^2$. The works in this section do not describe how $\|\boldsymbol{\omega}\|$ can converge to an equilibrium value that results in a fixed relative or rotational learning rate. In Weight Normalization, Salimans and Kingma (2016) point out how normalization can make parameters scale-invariant and that the gradient magnitude varies based on the weight magnitude. They describe how the gradient can "self-stabilize its norm", with larger gradients becoming smaller over time due to growth in the weight magnitude, but do not consider the effects of weight decay on this process. Zhang et al. (2019) and Hoffer et al. (2018) empirically find that the regularization effects of weight decay are primarily caused by increases in the effective learning rate due to decreased weight norms. Li and Arora (2020) show that weight decay can be replaced by an exponentially increasing learning rate when optimizing scale-invariant weights with SGDM.

### A.3    EQUILIBRIUM

These works also consider the fact that the weight norm converges to a specific value and the resulting effects on the relative update size. Van Laarhoven (2017) points out the scale-invariance property of normalization and how it interacts with L2 regularization. They derive the $\eta/\|\boldsymbol{\omega}\|^2$ as the effective learning rate and also how there exists a fixed point where the weight norms are stable. Their work does not consider convergence of the weight magnitude as a separate process from the overall convergence of the loss and weights. In Online Normalization, Chiley et al. (2019) show a simple derivation of the equilibrium condition in SGD and how it results in a relative update size that is identical across layers. The Spherical Motion Dynamics (SMD) (Wan et al., 2021) expands on prior work by proving the convergence of the weight norm and extending the analysis to include momentum. They also show plots of the weight norm over the course of training, providing empirical evidence for early convergence of the weight norm and also how it can fall out of equilibrium with sudden learning rate changes or when the learning rate becomes too small. They also consider the angular updates, empirically and analytically showing that they converge to an equilibrium value. Li et al. (2020) analyze the convergence of SGD to equilibrium by modelling it as a stochastic differential equation, arriving at similar conclusion as the SMD paper (without momentum). This is expanded upon by Li et al. (2022b).

### A.4    PROJECTED OPTIMIZATION

Some existing works are based on projections onto a sphere but do not scale the update to be proportional to the weight magnitude. Although similar to our Rotational Variants, this has a very different effect. Instead of rotating all scale-invariant groups at the same rate, they are kept at different rates based on their magnitude and resulting gradient norm where applicable. AdamP (Heo et al., 2021) orthogonalizes the update of the Adam and SGDM optimizers by removing the radial component of $\Delta_g\boldsymbol{\omega}$. The main reason for this is to avoid a rapid increase the weight norm during the initial phases of training. However, they use low amounts of weight decay if any which can also prevent the norms from growing as we have shown. Our Rotational Optimizer Variants keep the relative learning rate at a fixed level and do not suffer from the effect they report. Zhou et al. (2021) propose keeping the weight magnitude constant, projecting it onto a sphere after every step and removing the weight decay. Kodryan et al. (2022) analyze training using projections onto the unit sphere after every optimizer update. They consider the union of all scale-invariant weights and do not normalize their step sizes so their effective learning rate per group can vary, giving dynamics

that are generally quite different from SMD equilibrium. Roburin et al. (2020) analyzes the spherical projection of the Adam optimization trajectory during standard training.

## A.5 RELATIVE OPTIMIZATION

LARS (You et al., 2017) and LAMB (You et al., 2020) are variants of SGDM and AdamW that scale the update of a weight to be proportional to its norm (sometimes a clamped version of the weight norm). They apply this to linear and convolutional layer weights, keeping the original update for weights and biases. LARS and LAMB were proposed for large batch size training and found to work well there. Although they are not inspired by the Spherical Motion Dynamics, their form is quite similar to the Rotation Wrapper (Algorithm 1) with a few important distinctions. The default form of our Rotational Optimizer Variants (RVs) is applied filter-wise, centers the weights and allows the update magnitude to vary between steps while keeping the average relative update constant. The RV also doesn't apply weight decay while LARS and LAMB consider it a part of the update and take into account when scaling the relative update. Finally the RVs adjust the learning rate based on the SMD equilibrium value. This makes it more compatible with the underlying optimizer variants in terms of hyperparameters. One notable difference is the square root dependency on the relative updates in the SMD, while LARS and LAMB are directly proportional. This means that any learning rate schedule for these optimizers is more similar to applying a squared version of this schedule to standard optimizers or the RVs. This does not fully resolve the differences however, because changing the schedule also affects gains and biases where the update magnitude is directly proportional to the learning rate for all the optimizers and variants discussed here.

Nero (Liu et al., 2021) is another optimizer that applies relative updates that are directly proportional to the learning rate and weight magnitude. Like LARS and LAMB, Nero is not inspired by the SMD and to the best of our knowledge their relationship has not been pointed out before. Like the RVs, Nero is applied filter-wise and centers the weights. Overall, Nero is similar to the SGDM RV without momentum and the hyperparameter mapping, but also applies Adam like updates to the gains and biases, using a separate learning rate. By making the relative updates directly proportional to the learning rate, it has the same learning rate scheduling differences as LARS and LAMB mentioned above. Nero lacks momentum which is something that we observed can hurt large batch size training (exploratory experiments not shown).

Instead of controlling the average relative update size, Brock et al. (2021b) and Li et al. (2022a) clip the maximum relative update size instead. The Adaptive Gradient Clipping from Brock et al. (2021b) is applied on a per filter basis and is constant throughout training, i.e. does not vary with the learning rate or weight decay. The clipping introduced in Li et al. (2022a) scales with the learning rate and weight decay in a manner inspired by the equilibrium norm for SGD. They seem to apply this globally (i.e., not per neuron or layer).

## B  NORMALIZATION AND SCALE-INVARIANCE

**Setup:** We use Batch Normalization (Ioffe and Szegedy, 2015) as an example of a normalization operation. Let $\boldsymbol{x} = \boldsymbol{Z}\boldsymbol{\omega}$ for $\boldsymbol{x} \in \mathbb{R}^{B \times 1}$, $\boldsymbol{\omega} \in \mathbb{R}^{C \times 1}$ and $\boldsymbol{Z} \in \mathbb{R}^{B \times C}$ correspond to a single output feature of a linear layer (i.e. a neuron). We can write the batch normalization of this feature as:

$$\hat{\boldsymbol{x}} = N(\boldsymbol{x}) = \frac{\boldsymbol{x} - \mu}{\sqrt{\sigma^2 + \varepsilon}}, \qquad \mu = \frac{1}{B}\sum_{i=1}^{B} x_i, \qquad \sigma^2 = \frac{1}{B}\sum_{i=1}^{B}(x_i - \mu)^2 \qquad (19)$$

where $\boldsymbol{x} = [x_1, \ldots, x_B]^\top \in \mathbb{R}^B$ is a vector and $\varepsilon \geq 0$ is a small hyperparameter added for numerical stability. Backpropagation accurately treats $\mu$ and $\sigma$ as functions of $\boldsymbol{x}$. When $\varepsilon$ is sufficiently small to be ignored, the output of the normalization is not affected by a positive scaling of the input:

$$N(r\boldsymbol{x}) = (r\boldsymbol{x} - r\mu)/\sqrt{r^2\sigma^2 + \varepsilon} \approx (\boldsymbol{x} - \mu)/\sqrt{\sigma^2 + \varepsilon} = N(\boldsymbol{x}), \qquad r > 0 \qquad (20)$$

If the training loss $\mathscr{L}$ does not depend on $\boldsymbol{x}$ in other ways than through $N(\boldsymbol{x})$, this makes $\boldsymbol{x}$ scale-invariant with respect to the loss, i.e. $\mathscr{L}(r\boldsymbol{\omega}) = \mathscr{L}(\boldsymbol{\omega})$ for $r > 0$. Note that although we sometimes write $\mathscr{L}(\boldsymbol{\omega})$ for brevity the loss generally depends on other weights and inputs as well, $\boldsymbol{\omega}$ is generally only a portion of the parameters used in the network, and could for example be a particular row in the weight matrix of a fully connected layer. Some normalization operations like Centered Weight

Normlization (Huang et al., 2017) a.k.a. Weight Standardization (Qiao et al., 2019) are performed directly on the weights instead of activations. This also makes the weight scale-invariant and in case of the aforementioned methods also makes $\nabla_{\boldsymbol{\omega}}\mathscr{L}(\boldsymbol{\omega}) \perp \mathbf{1}$.

**Properties:** Scale-invariance results in the properties stated in Equations (1) and (2), repeated below:

$$\text{Gradient orthogonality:} \quad \nabla_{\boldsymbol{\omega}}\mathscr{L}(\boldsymbol{\omega}) \perp \boldsymbol{\omega} \tag{21}$$

$$\text{Inverse proportionality:} \quad \|\nabla_{\boldsymbol{\omega}}\mathscr{L}(\boldsymbol{\omega})\| \propto \|\boldsymbol{\omega}\|^{-1} \tag{22}$$

**Intuition:** The first property is a result of the loss surface being invariant along the direction of $\boldsymbol{\omega}$. Hence the directional derivative of $\mathscr{L}(\boldsymbol{\omega})$ in the direction of $\boldsymbol{\omega}$ is zero:

$$\langle \nabla_{\boldsymbol{\omega}}\mathscr{L}(\boldsymbol{\omega}), \frac{\boldsymbol{\omega}}{\|\boldsymbol{\omega}\|}\rangle = \lim_{h \to 0} \frac{\mathscr{L}(\boldsymbol{\omega} + h\boldsymbol{\omega}/\|\boldsymbol{\omega}\|) - \mathscr{L}(\boldsymbol{\omega})}{h} \tag{23}$$

$$= \lim_{h \to 0} \frac{\mathscr{L}(\boldsymbol{\omega}) - \mathscr{L}(\boldsymbol{\omega})}{h} \tag{24}$$

$$= \lim_{h \to 0} \frac{0}{h} = 0 \tag{25}$$

The second property is a result of the backpropagation through $N$, which scales the gradient by the factor used on the forward pass $1/\sqrt{\sigma^2 + \varepsilon} \approx \sigma^{-1}$ as if it were a constant, and the fact that $\sigma \propto \|\boldsymbol{\omega}\|$.

**Backpropagation:** The properties can also be shown using expressions for the backpropagation through the normalization layers. For completeness we include the learnable affine transformation that typically follows normalization operations:

$$\boldsymbol{y} = \gamma \hat{\boldsymbol{x}} + \beta \tag{26}$$

For the backpropagation we have:

$$\nabla_{\gamma}\mathscr{L}(\boldsymbol{p}) = \langle \hat{\boldsymbol{x}}, \nabla_{\boldsymbol{y}}\mathscr{L}(\boldsymbol{p})\rangle \tag{27}$$

$$\nabla_{\beta}\mathscr{L}(\boldsymbol{p}) = \langle \mathbf{1}_B, \nabla_{\boldsymbol{y}}\mathscr{L}(\boldsymbol{p})\rangle \tag{28}$$

$$\nabla_{\boldsymbol{x}}\mathscr{L}(\boldsymbol{p}) = \frac{\gamma}{\sqrt{\sigma^2+\varepsilon}} \cdot \left[ \nabla_{\boldsymbol{y}}\mathscr{L}(\boldsymbol{p}) - \tfrac{1}{B}\langle \mathbf{1}_B, \nabla_{\boldsymbol{y}}\mathscr{L}(\boldsymbol{p})\rangle \mathbf{1}_B - \tfrac{1}{B}\langle \hat{\boldsymbol{x}}, \nabla_{\boldsymbol{y}}\mathscr{L}(\boldsymbol{p})\rangle \hat{\boldsymbol{x}} \right] \tag{29}$$

Assuming that $\varepsilon$ is small gives:

$$\nabla_{\boldsymbol{x}}\mathscr{L}(\boldsymbol{p}) = \frac{\gamma}{\sigma} \left[ \nabla_{\boldsymbol{y}}\mathscr{L}(\boldsymbol{p}) - \tfrac{1}{B}\langle \mathbf{1}_B, \nabla_{\boldsymbol{y}}\mathscr{L}(\boldsymbol{p})\rangle \mathbf{1}_B - \tfrac{1}{B}\langle \hat{\boldsymbol{x}}, \nabla_{\boldsymbol{y}}\mathscr{L}(\boldsymbol{p})\rangle \hat{\boldsymbol{x}} \right] \tag{30}$$

In this case we have:

$$\langle \nabla_{\boldsymbol{x}}\mathscr{L}(\boldsymbol{p}), \mathbf{1}_B\rangle = \frac{\gamma}{\sigma} \left[ \langle \mathbf{1}_B, \nabla_{\boldsymbol{y}}\mathscr{L}(\boldsymbol{p})\rangle - \tfrac{1}{B}\langle \mathbf{1}_B, \nabla_{\boldsymbol{y}}\mathscr{L}(\boldsymbol{p})\rangle \underbrace{\langle \mathbf{1}_B, \mathbf{1}_B\rangle}_{=B} - \tfrac{1}{B}\langle \hat{\boldsymbol{x}}, \nabla_{\boldsymbol{y}}\mathscr{L}(\boldsymbol{p})\rangle \underbrace{\langle \hat{\boldsymbol{x}}, \mathbf{1}_B\rangle}_{=0} \right]$$
$$= 0 \tag{31}$$

and similarly:

$$\langle \nabla_{\boldsymbol{x}}\mathscr{L}(\boldsymbol{p}), \hat{\boldsymbol{x}}\rangle = \frac{\gamma}{\sigma} \left[ \langle \hat{\boldsymbol{x}}, \nabla_{\boldsymbol{y}}\mathscr{L}(\boldsymbol{p})\rangle - \tfrac{1}{B}\langle \mathbf{1}_B, \nabla_{\boldsymbol{y}}\mathscr{L}(\boldsymbol{p})\rangle \underbrace{\langle \mathbf{1}_B, \hat{\boldsymbol{x}}\rangle}_{=0} - \tfrac{1}{B}\langle \hat{\boldsymbol{x}}, \nabla_{\boldsymbol{y}}\mathscr{L}(\boldsymbol{p})\rangle \underbrace{\langle \hat{\boldsymbol{x}}, \hat{\boldsymbol{x}}\rangle}_{=B} \right]$$
$$= 0 \tag{32}$$

which gives:

$$\langle \nabla_{\boldsymbol{x}}\mathscr{L}(\boldsymbol{p}), \boldsymbol{x}\rangle = \langle \nabla_{\boldsymbol{x}}\mathscr{L}(\boldsymbol{p}), \sigma\hat{\boldsymbol{x}} + \mu\mathbf{1}_B\rangle = 0 \tag{33}$$

This allows us to obtain the properties of the weight gradient:

$$\nabla_{\boldsymbol{p}}\mathscr{L}(\boldsymbol{p}) = \boldsymbol{Z}^{\top}\nabla_{\boldsymbol{x}}\mathscr{L}(\boldsymbol{p}) \tag{34}$$

First we note that:

$$\|\nabla_{\boldsymbol{p}}\mathscr{L}(\boldsymbol{p})\| \propto \|\nabla_{\boldsymbol{x}}\mathscr{L}(\boldsymbol{p})\| \propto \sigma^{-1} \propto \|\boldsymbol{p}\|^{-1} \tag{35}$$

where the second proportionality follows from (30) and the final one from (19). This gives the inverse proportionality listed in Equation (22).

We can also derive the gradient orthogonality in Equation (21) as follows:

$$\langle \nabla_{\boldsymbol{p}}\mathscr{L}(\boldsymbol{p}), \boldsymbol{p} \rangle = \langle \boldsymbol{Z}^\top \nabla_{\boldsymbol{x}}\mathscr{L}(\boldsymbol{p}), \boldsymbol{p} \rangle \tag{36}$$
$$= \boldsymbol{p}^\top \boldsymbol{Z}^\top \nabla_{\boldsymbol{x}}\mathscr{L}(\boldsymbol{p}) \tag{37}$$
$$= \boldsymbol{x}^\top \nabla_{\boldsymbol{x}}\mathscr{L}(\boldsymbol{p}) \tag{38}$$
$$= \langle \nabla_{\boldsymbol{x}}\mathscr{L}(\boldsymbol{p}), \boldsymbol{x} \rangle \tag{39}$$
$$= 0 \tag{40}$$

These properties can also be shown directly from the scale-invariance using calculus theorems as done in Wan et al. (2021).

## C  SGDM Equilibrium

The standard version of SGD with momentum (SGDM) can be written as:

$$\boldsymbol{m}_t = \alpha \boldsymbol{m}_{t-1} + \boldsymbol{g}_t + \lambda \boldsymbol{p}_{t-1} \tag{41}$$
$$\boldsymbol{p}_t = \boldsymbol{p}_{t-1} - \eta \cdot \boldsymbol{m}_t \tag{42}$$

Where $\boldsymbol{p}_t \in \mathbb{R}^C$ is a parameter vector at time $t$, $\boldsymbol{g}_t = \nabla_{\boldsymbol{p}}\mathscr{L}(\boldsymbol{p}_t)$ is the gradient, $\boldsymbol{m}$ is the first moment and $\boldsymbol{v}$ is the second moment. The learning rate ($\eta \geq 0$), weight decay ($\lambda \geq 0$), momentum coefficient ($0 < \alpha < 1$) are hyperparameters.

We compute the total weight change due to $\boldsymbol{g}_t$, i.e. $\boldsymbol{u}$ in Equation (6) as:

$$\boldsymbol{u} = -\eta \sum_{k=t}^{\infty} \alpha^{k-t} \boldsymbol{g}_t = \tfrac{-\eta}{1-\alpha} \boldsymbol{g}_t \tag{43}$$

Analogously, the total weight change due to $\boldsymbol{w}_t$, i.e. $\boldsymbol{d}$ in Equation (6) is:

$$\boldsymbol{d} = -\eta \sum_{k=t}^{\infty} \alpha^{k-t} \lambda \boldsymbol{p}_t = \tfrac{-\eta\lambda}{1-\alpha} \boldsymbol{p}_t \tag{44}$$

Combining (43) and (44), this allows us to solve (6) for a scale-invariant weight vector $\boldsymbol{\omega}$. Here we assume scale-invariance since it slightly changes the resulting expression due to the dependency of $\|\boldsymbol{u}\|$ on $\|\boldsymbol{\omega}\|$. It also simplifies the math a bit, with $\boldsymbol{u} \perp \boldsymbol{\omega}$, not just in expectation. We get:

$$\widehat{\|\boldsymbol{\omega}\|}^2 = \mathbb{E}\left[ (\widehat{\|\boldsymbol{\omega}\|} - \|\boldsymbol{d}\|)^2 + \|\boldsymbol{u}\|^2 \right] \tag{45}$$
$$= \mathbb{E}\left[ \left( \widehat{\|\boldsymbol{\omega}\|} - \tfrac{\eta\lambda}{1-\alpha}\widehat{\|\boldsymbol{\omega}\|} \right)^2 + \left( \tfrac{\eta}{1-\alpha} \tfrac{\|\tilde{\boldsymbol{g}}\|}{\widehat{\|\boldsymbol{\omega}\|}} \right)^2 \right] \tag{46}$$

Where we define $\tilde{\boldsymbol{g}}_t = \boldsymbol{g}_t \|\boldsymbol{\omega}_t\|$ using $\|\boldsymbol{g}_t\| \propto \|\boldsymbol{\omega}_t\|^{-1}$ due to the inverse proportionality of the gradient magnitude, see Equation (2) or (22). We can interpret $\tilde{\boldsymbol{g}}_t$ as the gradient for weights of unit norm $\|\boldsymbol{\omega}_t\| = 1$.

Solving for $\widehat{\|\boldsymbol{\omega}\|}$ and assuming that $\eta\lambda \ll 2 \cdot (1-\alpha)$ gives:

$$\widehat{\|\boldsymbol{\omega}\|} = \sqrt[4]{\frac{\eta\mathbb{E}[\|\tilde{\boldsymbol{g}}\|^2]}{2\lambda \cdot (1-\alpha) - \eta\lambda^2}} \approx \sqrt[4]{\frac{\eta\mathbb{E}[\|\tilde{\boldsymbol{g}}\|^2]}{2\lambda \cdot (1-\alpha)}} \tag{47}$$

To obtain the absolute size of an update, we further assume that $\mathbb{E}[\|\boldsymbol{g}_t\|^2]$ can be approximated as a constant $\mathbb{E}[\|\boldsymbol{g}\|^2]$ when computing the size of $\boldsymbol{m}_t$, and that successive gradients are roughly orthogonal giving $\boldsymbol{m}_{t-1} \perp \boldsymbol{g}_t$ in expectation. For the random walk setting, the first is reasonable when the norm is stable e.g. around equilibrium and the second always holds. The average square

size of an update is then:

$$\mathbb{E}[\|\Delta_g \boldsymbol{p}\|^2] = \eta^2 \mathbb{E}\left[\|\alpha \boldsymbol{m}_{t-1} + \boldsymbol{g}_t\|^2\right] \tag{48}$$

$$= \eta^2 \mathbb{E}\left[\|\alpha \boldsymbol{m}_{t-1}\|^2\right] + \mathbb{E}\left[\|\boldsymbol{g}_t\|^2\right] \tag{49}$$

$$= \eta^2 \sum_{k=0}^{t} \mathbb{E}[(\alpha^{t-k}\|\boldsymbol{g}_k\|)^2] \tag{50}$$

$$\approx \eta^2 \frac{\mathbb{E}[\|\boldsymbol{g}\|^2]}{1-\alpha^2} \tag{51}$$

where (49) comes from the orthogonality, (50) by recursively writing out $\boldsymbol{m}$ in terms of $\boldsymbol{g}$, and (51) from assuming that $t$ is high enough to approximate the sum of the geometric series as an infinite sum.

Simplifying, we get the $\eta_g = \sqrt{\mathbb{E}[\|\Delta_g \boldsymbol{p}\|^2]}$ and $\eta_r = \sqrt{\mathbb{E}[\|\Delta_g \boldsymbol{\omega}\|^2]}/\widehat{\|\boldsymbol{\omega}\|}$ in Table 1. We note that the derived rates are consistent with the ones derived in the Spherical Motion Dynamics (Wan et al., 2021) and Online Normalization (Chiley et al., 2019) (when $\alpha = 0$).

## D  LION EQUILIBRIUM

The standard version of Lion (Chen et al., 2023) can be written as:

$$\boldsymbol{v}_t = \text{sign}(\beta_1 \boldsymbol{m}_{t-1} + (1 - \beta_1)\boldsymbol{g}_t) \tag{52}$$

$$\boldsymbol{m}_t = \beta_2 \boldsymbol{m}_{t-1} + (1 - \beta_2)\boldsymbol{g}_t \tag{53}$$

$$\boldsymbol{p}_t = \boldsymbol{p}_{t-1} - \eta \cdot (\boldsymbol{v}_t + \lambda \boldsymbol{p}_{t-1}) \tag{54}$$

Where $\boldsymbol{p}_t \in \mathbb{R}^C$ is a parameter vector at time $t$, $\boldsymbol{g}_t = \nabla_{\boldsymbol{p}}\mathcal{L}(\boldsymbol{p}_t)$ is the gradient, $\boldsymbol{m}$ is the first moment and $\boldsymbol{v}$ is the update velocity. The learning rate ($\eta \geq 0$), weight decay ($\lambda \geq 0$), moment coefficients ($0 < \beta_1 < 1, 0 < \beta_2 < 1$).

In our analysis we look at the arguments of the sign function which we define as:

$$\boldsymbol{n}_t := \beta_1 \boldsymbol{m}_{t-1} + (1 - \beta_1)\boldsymbol{g}_t, \qquad \boldsymbol{v}_t = \text{sign}(\boldsymbol{n}_t) \tag{55}$$

To obtain an estimate of the magnitude $\|\boldsymbol{n}_t\|$, we assume that the gradient magnitude $\mathbb{E}[\|\boldsymbol{g}_t\|^2]$ can be approximated as a constant $\mathbb{E}[\|\boldsymbol{g}\|^2]$ and that successive gradients are roughly orthogonal giving $\boldsymbol{m}_{t-1} \perp \boldsymbol{g}_t$ in expectation. For the random walk setting, the first is reasonable when the norm is stable e.g. around equilibrium and the second always holds. This gives:

$$\mathbb{E}[\|\boldsymbol{n}_t\|^2] = \mathbb{E}\left[\|\beta_1 \boldsymbol{m}_{t-1} + (1 - \beta_1)\boldsymbol{g}_t\|^2\right] \tag{56}$$

$$= \beta_1^2 \mathbb{E}\left[\|\beta_2 \boldsymbol{m}_{t-1} + (1 - \beta_2)\boldsymbol{g}_{t-1}\|^2\right] + (1 - \beta_1)^2 \mathbb{E}\left[\|\boldsymbol{g}_t\|^2\right] \tag{57}$$

$$= \beta_1^2 (1 - \beta_2)^2 \sum_{k=0}^{k=t-1} \beta_2^{t-1-k} \mathbb{E}[\|\boldsymbol{g}_k\|^2] + (1 - \beta_1)^2 \mathbb{E}\left[\|\boldsymbol{g}_t\|^2\right] \tag{58}$$

$$\approx \beta_1^2 (1 - \beta_2)^2 \sum_{k=0}^{\infty} \beta_2^{k} \mathbb{E}[\|\boldsymbol{g}\|^2] + (1 - \beta_1)^2 \mathbb{E}\left[\|\boldsymbol{g}\|^2\right] \tag{59}$$

$$= \left((1 - \beta_1)^2 + \beta_1^2 \frac{1 - \beta_2}{1 + \beta_2}\right) \mathbb{E}[\|\boldsymbol{g}\|^2] \tag{60}$$

where we have used the gradient orthogonality and constant magnitude and approximated the geometric sum as extending to infinity.

To compute the gradient contribution $\|\boldsymbol{u}\|$ in Equation (6), we first need to model how the sign non-linearity affects the magnitude and direction of the update. We note that for $\boldsymbol{p}_t \in \mathbb{R}^C$:

$$\|\boldsymbol{v}_t\| = \sqrt{C} \tag{61}$$

so the sign function has an average scaling effect:

$$\frac{\|\boldsymbol{v}_t\|}{\|\boldsymbol{n}_t\|} = \sqrt{\frac{C}{\left((1 - \beta_1)^2 + \beta_1^2 \frac{1-\beta_2}{1+\beta_2}\right) \mathbb{E}[\|\boldsymbol{g}\|^2]}} \tag{62}$$

The sign function will also rotate $\boldsymbol{n}_t$ resulting in two components, one parallel to $\boldsymbol{n}_t$ and the other orthogonal. We will assume that the orthogonal one cancels out on average without significantly affecting equilibrium and focus on the parallel component. This component depends on the average angle between $\boldsymbol{n}_t$ and $\text{sign}(\boldsymbol{n}_t)$ which is determined by the distribution and correlation between the elements. In the random walk setting, we can assume the components of $\boldsymbol{n}_t = [n_1, \ldots, n_C]$ are normally distributed with mean zero. However, the expression for the average angle is still complicated unless the components are independent and identically distributed (i.i.d.) so we make this assumption for this step with $n_k \sim \mathcal{N}(0, \sigma^2)$ i.i.d. for all $k$. Then we can use the known expected absolute value for a centered normal distribution to get:

$$\mathbb{E}[\langle \boldsymbol{n}_t, \text{sign}(\boldsymbol{n}_t)\rangle] = C \cdot \mathbb{E}[|n_k|] = C \cdot \sqrt{\frac{2\sigma^2}{\pi}} = \sqrt{\frac{2}{\pi}} \cdot \|\boldsymbol{n}_t\| \cdot \|\text{sign}(\boldsymbol{n}_t)\| \tag{63}$$

Note that the angle is still bounded regardless of the distribution but will result in a different factor in the range that $\|\boldsymbol{n}\|_1/(\sqrt{C}\|\boldsymbol{n}\|_2)$ can take, i.e. $[C^{-\frac{1}{2}}, 1]$ instead of $\sqrt{2/\pi}$.

Based on the preceding analysis we will model the sign function for the computation of $\|\boldsymbol{u}\|$ as:

$$\text{sign}(\boldsymbol{n}_t) \approx \sqrt{\frac{2}{\pi}} \frac{\|\boldsymbol{v}_t\|}{\|\boldsymbol{n}_t\|} \boldsymbol{n}_t = \sqrt{\frac{2C}{\mathbb{E}[\|\boldsymbol{g}\|^2] \cdot \pi \cdot \left((1-\beta_1)^2 + \beta_1^2 \frac{1-\beta_2}{1+\beta_2}\right)}} \boldsymbol{n}_t \tag{64}$$

which gives:

$$\mathbb{E}[\|\boldsymbol{u}\|^2] = \frac{2\eta C}{\mathbb{E}[\|\boldsymbol{g}\|^2] \cdot \pi \cdot \left((1-\beta_1)^2 + \beta_1^2 \frac{1-\beta_2}{1+\beta_2}\right)} \mathbb{E}\left[\left\|(1-\beta_1)\boldsymbol{g} + \beta_1(1-\beta_2)\sum_{k=0}^{\infty}\beta_2^k \boldsymbol{g}\right\|^2\right] \tag{65}$$

$$= \frac{2\eta C}{\pi \cdot \left((1-\beta_1)^2 + \beta_1^2 \frac{1-\beta_2}{1+\beta_2}\right)} \tag{66}$$

Combined with $\boldsymbol{d} = -\eta\lambda\boldsymbol{\omega}_{t-1}$, this allows us to solve (6) for a scale-invariant weight vector $\boldsymbol{\omega}$:

$$\widehat{\|\boldsymbol{\omega}\|}^2 = \mathbb{E}\left[(\widehat{\|\boldsymbol{\omega}\|} - \|\boldsymbol{d}\|)^2 + \|\boldsymbol{u}\|^2\right] \tag{67}$$

$$= (\widehat{\|\boldsymbol{\omega}\|} - \eta\lambda\widehat{\|\boldsymbol{\omega}\|})^2 + \mathbb{E}[\|\boldsymbol{u}\|^2] \tag{68}$$

$$= (1 - 2\eta\lambda + \eta^2\lambda^2)\widehat{\|\boldsymbol{\omega}\|}^2 + \frac{2\eta C}{\pi}\left((1-\beta_1)^2 + \beta_1^2\frac{1-\beta_2}{1+\beta_2}\right)^{-1} \tag{69}$$

Solving for $\widehat{\|\boldsymbol{\omega}\|}$ and assuming $\eta\lambda \ll 1$ gives:

$$\widehat{\|\boldsymbol{\omega}\|} = \sqrt{\frac{2\eta C}{\pi \cdot (2\lambda - \eta\lambda^2)}}\left((1-\beta_1)^2 + \beta_1^2\frac{1-\beta_2}{1+\beta_2}\right)^{-\frac{1}{2}} \approx \sqrt{\frac{\eta C}{\pi\lambda}}\left((1-\beta_1)^2 + \beta_1^2\frac{1-\beta_2}{1+\beta_2}\right)^{-\frac{1}{2}} \tag{70}$$

Combined with $\|\boldsymbol{v}_t\| = \sqrt{C}$ for $\boldsymbol{\omega}, \boldsymbol{p} \in \mathbb{R}^C$ we get:

$$\widehat{\eta}_g = \sqrt{\mathbb{E}[\|\Delta_g\boldsymbol{p}\|^2]} = \eta\sqrt{C} \tag{71}$$

$$\widehat{\eta}_r = \sqrt{\mathbb{E}[\|\Delta_g\boldsymbol{\omega}\|^2]}/\widehat{\|\boldsymbol{\omega}\|} = \sqrt{\pi\eta\lambda} \cdot \left((1-\beta_1)^2 + \beta_1^2\frac{1-\beta_2}{1+\beta_2}\right)^{\frac{1}{2}} \tag{72}$$

# E  ADAM+$L_2$ EQUILIBRIUM

In this section we apply a modified form of the geometric model from Section 3.1 to Adam (Kingma and Ba, 2015) with $L_2$ regularization (Adam+$L_2$ for short) to gain some insight into how the rotational equilibrium differs from that of Adam with decoupled weight decay (AdamW, see Section 3.2).

## E.1  ADAM+$L_2$ FORMULATION

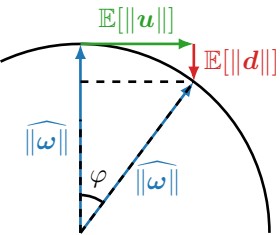

Figure 9: Weight norm equilibrium. The loss gradient causes an update $\boldsymbol{u}$ and the weight decay $\boldsymbol{d}$.

We will write the Adam+$L_2$ update as follows:

$$\boldsymbol{m}_t = \beta_1 \boldsymbol{m}_{t-1} + (1 - \beta_1)(\boldsymbol{g}_t + \lambda \boldsymbol{p}_{t-1}) \tag{73}$$

$$\boldsymbol{v}_t = \beta_2 \boldsymbol{v}_{t-1} + (1 - \beta_2)(\boldsymbol{g}_t + \lambda \boldsymbol{p}_{t-1})^2 \tag{74}$$

$$\boldsymbol{p}_t = \boldsymbol{p}_{t-1} - \eta \cdot \left( \frac{\boldsymbol{m}_t/(1-\beta_1^t)}{\sqrt{\boldsymbol{v}_t/(1-\beta_2^t)} + \varepsilon} \right) \tag{75}$$

Similar to AdamW, $\boldsymbol{p}_t \in \mathbb{R}^C$ is a parameter vector at time $t$, $\boldsymbol{g}_t = \nabla_{\boldsymbol{p}} \mathcal{L}(\boldsymbol{p}_t)$ is the gradient, and all operations (e.g. division, squaring) are performed elementwise. In Adam+L2, both the first and second moment of the gradient ($\boldsymbol{m}$ and $\boldsymbol{v}$) include an $L_2$ regularization term. This differs from AdamW (see Equation 9) where the $L_2$ regularization (or weight decay) does not affect $\boldsymbol{m}$ and $\boldsymbol{v}$. The learning rate ($\eta \geq 0$), $L_2$ regularization coefficient ($\lambda \geq 0$), moment coefficients ($0 < \beta_1 < 1, 0 < \beta_2 < 1$) and $\varepsilon \geq 0$ are hyperparameters similar to AdamW. Like before, we use $\boldsymbol{\omega}$ to denote a weight vector $\boldsymbol{p}$ that can form rotational equilibrium (e.g. dot product weights, not biases etc).

### E.2   SIMPLIFICATIONS

The rotational dynamics of Adam+$L_2$ are more complicated than those of AdamW. The main difference is that the strength of the "weight decay" is affected by the gradient norm. As we will see, this makes the equilibrium norm and angular update depend on the gradient magnitude. Furthermore, the weight decay can be scaled differently for each coordinate of the weight vector as the gradient distribution may vary between them. This complicates the analysis, forcing us to treat each coordinate separately.

Our analysis is based on the random walk setup introduced in Section 3 and described in Appendix G. We further make several assumptions and simplifying approximations that allow us to obtain simpler expressions for the cases of interest:

1. We assume the rotational equilibrium exists as a steady state where hyperparameters are fixed (not varying over time), the expected weight norm $\sqrt{\mathbb{E}[\|\boldsymbol{\omega}_t\|^2]} = \widehat{\|\boldsymbol{\omega}\|}$ is constant, and the second moment of the gradient $\mathbb{E}[\boldsymbol{g}_t^2]$ is constant over time. For simplicity we will drop the $t$ subscript.

2. We focus on the case where the weights are scale-invariant, defining $\tilde{\boldsymbol{g}} = \|\boldsymbol{\omega}\|\boldsymbol{g}$ as the gradient corresponding to a unit weight norm based on the inverse proportionality from Equation 2.

3. We will assume that $\varepsilon$ and the bias correction can be ignored, i.e. that $\varepsilon$, $\beta_1^t$ and $\beta_2^t$ are all effectively zero.

4. We will assume the second moment tracker $\boldsymbol{v}_t$ is dominated by the gradient component, i.e. that $\boldsymbol{g}^2 \gg \lambda^2 \boldsymbol{\omega}^2$, and that it perfectly tracks the expected value, i.e. that $\mathbb{E}[\boldsymbol{v}] = \mathbb{E}[\boldsymbol{g}^2]$. This is a non-trivial approximation based on the geometry of equilibrium when the angular updates are small. For a small $\varphi$ in Figure 9 we can can approximate:

$$\mathbb{E}[\|\boldsymbol{u}\|] = \widehat{\|\boldsymbol{\omega}\|} \cdot \tan(\varphi) \approx \widehat{\|\boldsymbol{\omega}\|} \cdot \varphi \tag{76}$$

$$\mathbb{E}[\|\boldsymbol{d}\|] = \widehat{\|\boldsymbol{\omega}\|} \cdot (1 - \cos(\varphi)) \approx \widehat{\|\boldsymbol{\omega}\|} \cdot \frac{\varphi^2}{2} \tag{77}$$

As a result $\mathbb{E}[\|\boldsymbol{u}\|] \gg \mathbb{E}[\|\boldsymbol{d}\|]$. For Adam+$L_2$ we have $\boldsymbol{u} = -\eta \boldsymbol{g}_t/\sqrt{\boldsymbol{v}}$ and $\boldsymbol{d} = -\eta\lambda\boldsymbol{\omega}_{t-1}/\sqrt{\boldsymbol{v}}$. As long as $\boldsymbol{v}$ is relatively homogeneous across coordinates, we therefore have $\mathbb{E}[\|\boldsymbol{g}\|] \gg \mathbb{E}[\lambda\|\boldsymbol{\omega}\|]$. We assume this holds roughly coordinate wise as well, giving $\boldsymbol{g}^2 \gg \lambda^2\boldsymbol{\omega}^2$. We note that this fourth assumption is not strictly necessary but significantly simplifies the resulting expressions, giving us an interpretable closed form solution.

### E.3 Equilibrium Norm

The total weight change due to the gradient $\boldsymbol{g}_t$, i.e. $\boldsymbol{u}$ in Equation (6) is given by:

$$\boldsymbol{u} = -\eta \sum_{k=t}^{\infty} \beta_1^{k-t}(1-\beta_1)\frac{\boldsymbol{g}_t}{\sqrt{\boldsymbol{v}}} = -\eta\frac{\boldsymbol{g}_t}{\sqrt{\boldsymbol{v}}} \tag{78}$$

Similarly the total change due to the weight decay is:

$$\boldsymbol{d} = -\eta \sum_{k=t}^{\infty} \beta_1^{k-t}(1-\beta_1)\frac{\lambda\boldsymbol{\omega}_{t-1}}{\sqrt{\boldsymbol{v}}} = -\eta\frac{\lambda\boldsymbol{\omega}_{t-1}}{\sqrt{\boldsymbol{v}}} \tag{79}$$

Due to the coordinate-wise differences in the weight decay we analyze a single element $\omega_k$ at coordinate $k$ in $\boldsymbol{\omega}$ with corresponding elements $d_k, u_k, g_k, v_k$ in $\boldsymbol{d}, \boldsymbol{u}, \boldsymbol{g}, \boldsymbol{v}$, respectively. Although the geometric model is not well defined coordinate-wise, we can still use the concept of orthogonality as defined for random variables. This gives us:

$$\mathbb{E}[\omega_k^2] = \mathbb{E}[(\omega_k - d_k)^2 + u_k^2] \tag{80}$$

$$= (1 - \frac{\eta\lambda}{\sqrt{v_k}})^2 \mathbb{E}[\omega_k^2] + \eta^2\frac{\mathbb{E}[g_k^2]}{v_k} \tag{81}$$

$$= (1 - \frac{2\eta\lambda}{\sqrt{v_k}} + \frac{\eta^2\lambda^2}{v_k})\mathbb{E}[\omega_k^2] + \eta^2(1 - \frac{\lambda^2\mathbb{E}[\omega_k^2]}{v_k}) \tag{82}$$

$$= (1 - \frac{2\eta\lambda}{\sqrt{v_k}})\mathbb{E}[\omega_k^2] + \eta^2 \tag{83}$$

where we have used the fact that $v_k = \mathbb{E}[g_k^2] + \lambda^2\mathbb{E}[\omega_k^2]$.

Since we are targeting the scale-invariant case (Assumption 2) we can write:

$$\mathbb{E}[g_k^2] = \mathbb{E}[\tilde{g}_k^2]\mathbb{E}[\|\boldsymbol{\omega}\|^2]^{-1} \tag{84}$$

where $\tilde{g}_k$ corresponds to an element of the unit norm weight gradient $\tilde{\boldsymbol{g}}$. Accordingly we can write:

$$v_k = \mathbb{E}[\tilde{g}_k^2]\mathbb{E}[\|\boldsymbol{\omega}\|^2]^{-1} + \lambda^2\mathbb{E}[\omega_k^2] \approx \mathbb{E}[\tilde{g}_k^2]\mathbb{E}[\|\boldsymbol{\omega}\|^2]^{-1} \tag{85}$$

where we used Assumption 4.

Plugging this form of $v$ into Equation (83), squaring and simplifying gives:

$$\mathbb{E}[\omega_k^2] = \frac{\eta}{2\lambda}\sqrt{\frac{\mathbb{E}[\tilde{g}_k^2]}{\mathbb{E}[\|\boldsymbol{\omega}\|^2]}} \tag{86}$$

We can now write an expression for the equilibrium norm:

$$\widehat{\|\boldsymbol{\omega}\|}^2 = \mathbb{E}[\|\boldsymbol{\omega}\|^2] = \sum_{k=1}^{C}\mathbb{E}[\omega_k^2] = \frac{\eta}{2\lambda}\sum_{k=1}^{C}\sqrt{\frac{\mathbb{E}[\tilde{g}_k^2]}{\mathbb{E}[\|\boldsymbol{\omega}\|^2]}} \tag{87}$$

which gives:

$$\widehat{\|\boldsymbol{\omega}\|} = \sqrt[3]{\frac{\eta}{2\lambda}\cdot\langle\mathbf{1}, \sqrt{\mathbb{E}[\tilde{\boldsymbol{g}}^2]}\rangle} \tag{88}$$

where $\langle\cdot,\cdot\rangle$ denotes an inner product. Note that when the elements of $\boldsymbol{g}$ have the same second moment, e.g. when they are identically distributed, we can write $\langle\mathbf{1}, \sqrt{\mathbb{E}[\tilde{\boldsymbol{g}}^2]}\rangle = \mathbb{E}[\|\tilde{\boldsymbol{g}}\|^2]$. Also note how this behavior differs from that of AdamW, here the equilibrium norm depends on the gradient magnitude. Finally we note that without scale-invariance we would get a square root instead of a cube root.

## E.4 Equilibrium Angular Update

To obtain the absolute size of an update for $\boldsymbol{\omega}$ in equilibrium, we use the fact that in the random walk successive gradients are orthogonal in expectation.

Similar to AdamW, we can then write the average square size of an update as:

$$\mathbb{E}[\|\Delta_g \boldsymbol{\omega}\|^2] = \eta^2 (1 - \beta_1)^2 \sum_{k=0}^{t} \beta_1^{2t-2k} \mathbb{E}\left[\left\|\frac{\boldsymbol{g}_k}{\boldsymbol{v}}\right\|^2\right] \tag{89}$$

$$\approx \eta^2 \frac{1-\beta_1}{1+\beta_1} C \tag{90}$$

where we approximated the geometric sum with its limit and used $\boldsymbol{v} \approx \mathbb{E}[\boldsymbol{g}^2]$ based on Assumption 4. Note that the use of Assumption 4 gives the same result as for AdamW.

We can then approximate the expected angular update in equilibrium as:

$$\widehat{\eta_r} = \frac{\sqrt{\mathbb{E}[\|\Delta_g \boldsymbol{\omega}\|^2]}}{\widehat{\|\boldsymbol{\omega}\|}} = \sqrt[3]{\frac{2\eta^2 \lambda}{\langle \mathbf{1}, \sqrt{\mathbb{E}[\tilde{\boldsymbol{g}}^2]} \rangle}} \sqrt{\frac{1-\beta_1}{1+\beta_1}} C \tag{91}$$

Note that the average angular update depends on the gradient magnitude unlike for other optimizers. Also note the different dependency on $\eta$ and $\lambda$, here the angular update depends on the product $\eta^2 \lambda$, not $\eta\lambda$ like for other optimizers. Finally there is an odd dependency on $C$ that is not present in the other optimizers. Without scale-invariance the first cube root would be replaced by a square root and the gradient dependency on $C$ would cancel the $C$ in the second root.

## F  Simple System for Random Walks

**Definition:** We define the simple system as:

$$f(\boldsymbol{X}) = \boldsymbol{\gamma}_{\text{out}} \odot N\big(\boldsymbol{W}(\boldsymbol{\gamma}_{\text{in}} \odot \boldsymbol{X})\big) \tag{92}$$

where $\boldsymbol{X} \in \mathbb{R}^{C \times B}$, $\boldsymbol{W} \in \mathbb{R}^{K \times C}$, $\boldsymbol{\gamma}_{\text{in}} \in \mathbb{R}^{C \times 1}$, $\boldsymbol{\gamma}_{\text{out}} \in \mathbb{R}^{K \times 1}$ and $N$ is a batch normalization function (see Equation 19) applied to each feature independently. The only learnable parameters are the weights $\boldsymbol{W}$, the gammas $\boldsymbol{\gamma}_{\text{in}}$ and $\boldsymbol{\gamma}_{\text{out}}$ are kept constant. We initialize the weights using the default initialization for a linear layer in PyTorch (Paszke et al., 2019) i.e. each element is sampled independently and uniformly from the interval $[-\frac{1}{\sqrt{C}}, \frac{1}{\sqrt{C}}]$. The gammas are initialized with elements independent and identically distributed (i.i.d.) following a standard normal distribution. The inputs are also sampled i.i.d. from a standard normal distribution at each iteration. The gradients of $f(\boldsymbol{X})$, which are used to compute other gradients via the chain-rule or backpropagation, are sampled i.i.d. from a normal distribution with standard deviation $\frac{1}{KB}$ where the $B$ simulates the typical averaging of the loss over a batch and the $K$ gives a scale more similar to the derivatives of softmax-cross-entropy (the difference of two vectors with an $L_1$-norm of 1 each). We can also scale the initial gradients further with a loss scale to obtain different gradient norms (especially important for Adam).

**Rationale:** We use this system to study a random walk in a neural network as described in Section 3, which serves as a simplified model of a real optimization problem. The gammas give different variances for each input and output channel, causing the second gradient moment in Adam/AdamW to vary between elements of $\boldsymbol{W}$ like they may in real neural network training due to the rest of the network. The normalization ensures that the system is scale-invariant for each row of $\boldsymbol{W}$. The randomly sampled inputs and initial gradients ensure that everything is orthogonal in expectation. Compared to a real neural network training, the dynamics of this system are simplified with no loss converging over time and steady input / gradient distributions. Other complicated effects such as dead ReLUs do also not happen in this system. This makes this simple system a good setting to study the equilibrium dynamics in a controlled manner.

**Details of Figure** 4: Here we use $B = 32, C = K = 128$. We use the default hyperparameters for Adam and AdamW from PyTorch 2.0 (Paszke et al., 2019), except for the $L_2$-regularization coefficient of Adam which is zero by default but we use $\lambda = 10^{-3}$. For reference the other values are: learning rate $\eta = 10^{-3}$, first moment coefficient $\beta_1 = 0.9$, second moment coefficient $\beta_2 = 0.999$, epsilon $\varepsilon = 10^{-8}$, weight decay $\lambda = 10^{-1}$ (AdamW only). We do not use an additional loss scale for to scale the gradient norms. The experiments run for 30k steps, the plots are downsampled by a factor of 100x without averaging.

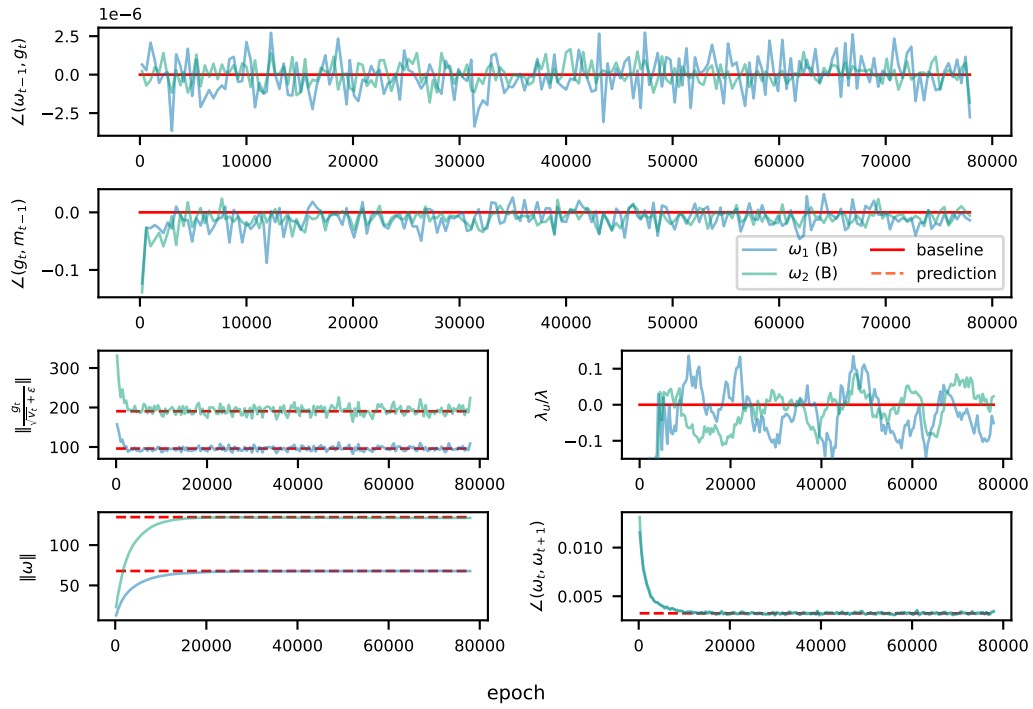

Figure 10: Measurement how closely the random walk model approximates the training dynamics of two convolutional filters of a ResNet-20 trained on CIFAR-10 with a constant learning rate.

## G   DIFFERENCES BETWEEN REAL NETWORKS AND A RANDOM WALK

The random walk model we use in our analysis lets us make a variety of simplifications that would not strictly hold for real neural network optimization problems. For the real networks we therefore effectively make several approximations that can affect the accuracy of the predictions. In this section, we begin by discussing the random walk model introduced in Section 3. Subsequently, we present measurements to evaluate the accuracy of the approximations and outline the differences for real neural networks.

**Random Walk Setup:** The random walk setup models training dynamics, where the batch gradient is dominated by the noise component. Of course this does not hold exactly in practice but may be a reasonable approximation for stochastic gradient descent with sufficiently small mini-batches. In this case the equilibrium dynamics may closely approximate that of the random walk.

In Section 3, we established our main simplification for the random walk setup: $g_{\mathbb{B}} = g_{\mathbb{X}} + g_N \approx g_N$ with $\mathbb{E}_{\mathbb{B}}[g_N] = 0$. In practice, this setup can be simulated by independently sampling inputs and gradients for the network outputs at each step from a zero-mean normal distribution. The parameter gradients are computed via backpropagation, i.e. by using the chain rule starting from the randomly sampled output gradients. Due to the linearity of backpropagation, the gradient of each parameter coordinate will be a weighted sum of the output gradients. As a result these gradients remain zero-mean and are independent between steps, but they will accurately capture effects such as gradient orthogonality (Equation 1) and inverse proportionality (Equation 2). Additionally, because we are randomly sampling the inputs and gradients for the network outputs from the same distribution over time, the distribution of these gradients does not change over time. This is in contrast to standard optimization with a non-random loss, where we do not expect gradient independence between steps and the distribution may change over time in a non-trivial manner.

**Normalized Setup:** We present measurements how closely the random walk model approximates the training dynamics of an original ResNet-20 trained on CIFAR-10 with a constant learning rate of $\eta = 0.01$ and a weight decay of $\lambda = 0.01$ in Figure 10. This standard ResNet has its convolutional layers followed by Batch Normalization, ensuring that the network is well-normalized. Consequently, we expect the convolutional weights to be scale-invariant.

Consistent with the expectation for this network, the angle between the gradient $\boldsymbol{g}_t$ and the weights $\boldsymbol{\omega}_{t-1}$ is close to zero. This is evident from the first row. The second row suggests that $\forall j \neq k$: $\mathbb{E}[\langle \boldsymbol{g}_j, \boldsymbol{g}_k \rangle] = 0$, which in average holds in random walk scenarios, also roughly holds here. We use $\boldsymbol{m}_{t-1} \perp \boldsymbol{g}_t$ as a measurement for this. It gives us information about the orthogonality of $\boldsymbol{g}_t$ and the previous update directions.

In the third row, we assess the simplifications related to the scaled gradient $\tilde{\boldsymbol{u}}_t = \boldsymbol{g}_t/(\sqrt{\boldsymbol{v}} + \varepsilon)$, an approximation of $\boldsymbol{u}$ with constant $\boldsymbol{v}$. The left panel depicts how $\mathbb{E}[\|\tilde{\boldsymbol{u}}_t\|]$ evolves over time. Our observations indicate that it closely aligns with our approximation $\sqrt{C}$ in this setup.

We further measure the weight decay component of the scaled gradient $\lambda_u$ by projecting it on the weight vector $\boldsymbol{\omega}$, $\lambda_u = \langle \boldsymbol{\omega}, \tilde{\boldsymbol{u}}_t \rangle/\|\boldsymbol{\omega}\|^2$. We take this approach to relate the weight decay component of the scaled gradient with the weight decay denoted as $\lambda$. The right panel of the third row illustrates this measurement. Notably, the gradient's weight decay component is relatively small, staying roughly within 10% of the weight decay.

Finally, in the fourth row, we compare the observed weight norms $\|\boldsymbol{\omega}_i\|$ and angular updates $\eta_r$ with our predictions from Table 1. We find that the predictions closely match the measurements after the initial transient phase in this setup.

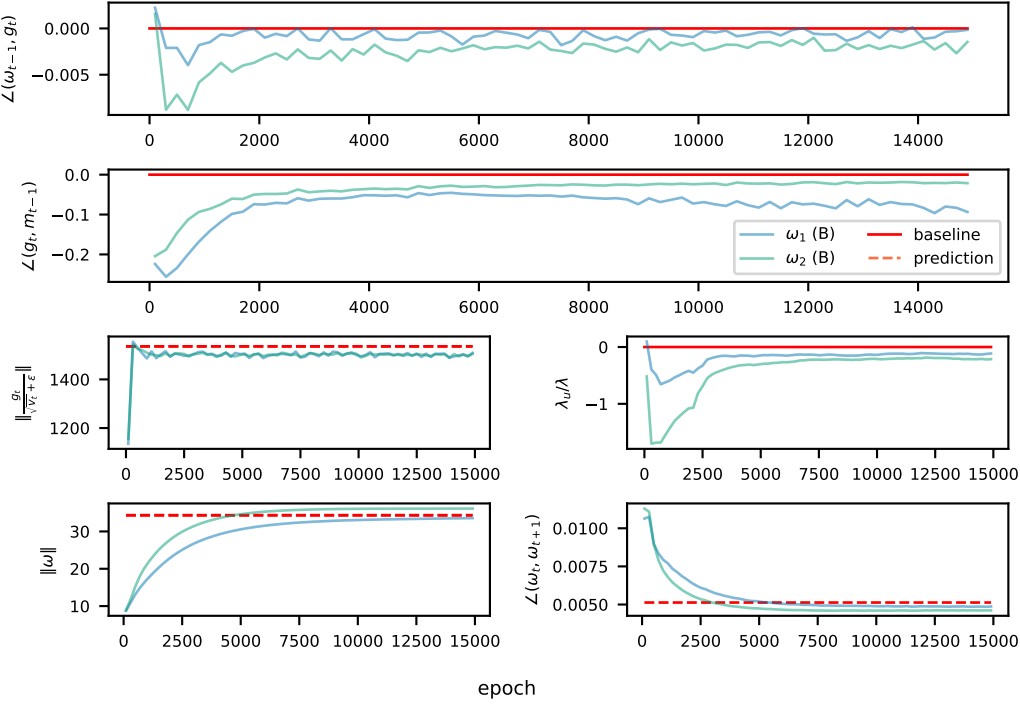

Figure 11: Measurement how closely the random walk model approximates the training dynamics of two linear layers for a GPT-like model trained on wikitext.

**Poorly Normalized Setup:** In this section, we evaluate how closely the random walk model approximates the training dynamics of a GPT-like model trained on Wikitext with learning rate $\eta = 0.0005$ and weight decay $\lambda = 0.5$ in Figure 11. The architecture does not incorporate Layer Normalization immediately after the linear layers. As a result, we do not expect the weight vectors $\boldsymbol{\omega}$ to be fully scale-invariant. The measurements in the first row supports this. For the angle between the gradient $\boldsymbol{g}_t$ and the weights $\boldsymbol{\omega}_{t-1}$ we measure a small bias in average opposed to the measurements of the normalized setup in Figure 10.

It is therefore not surprising that the scaled gradient $\tilde{\boldsymbol{u}}_t$, projected on the weight vector $\boldsymbol{\omega}$ has a more significant contribution in this setup, as evident in the right panel of the third row. As a consequence of the additional negative weight decay of the scaled gradient component—reducing the effective weight decay—our equilibrium norm prediction tends to underestimate the measured weight norm

$\|\boldsymbol{\omega}\|$ and over-estimate the expected angular update $\eta_r$. By defining the error as a scaling factor of $\lambda$ (represented as $\lambda_{err} = \frac{\lambda_e}{\lambda}$), we observe the following impact on our prediction

$$\eta_r \approx \sqrt{2\eta(\lambda \cdot \lambda_{err})\frac{1-\beta_1}{1+\beta_1}} = \widehat{\eta}_r \cdot \sqrt{\lambda_{err}} \tag{93}$$

$$\|\boldsymbol{\omega}\| \approx \sqrt{\frac{\eta C}{2(\lambda \cdot \lambda_{err})})} = \widehat{\|\boldsymbol{\omega}\|} \cdot \frac{1}{\sqrt{\lambda_{err}}} \tag{94}$$

At the same time we can see from the left panel in row three, that our prediction over-estimates the scaled gradient norm. This means we tend to over-estimate the equilibrium norm with our prediction, but not the expected angular update $\widehat{\eta}_r$. For $\widehat{\eta}_r$ the scaled gradient norm cancels out. For the equilibrium norm and error estimate $C_{err}$, $\|\tilde{\boldsymbol{g}}\| = C \cdot C_{err}$, we have:

$$\|\boldsymbol{\omega}\| \approx \sqrt{\frac{\eta(C \cdot C_{err})}{2\lambda}} = \widehat{\|\boldsymbol{\omega}\|} \cdot \sqrt{C_{err}} \tag{95}$$

Interestingly, we observe in the last row that for $\boldsymbol{\omega}_1$ these effects on the equilibrium weight norm seem to cancel out and our prediction $\widehat{\|\boldsymbol{\omega}\|}$ holds roughly for $\|\boldsymbol{\omega}_1\|$. For $\boldsymbol{\omega}_2$ we do under-estimate the equilibrium norm but note that the error is only a few percent.

Finally, the second row suggests that the approximation $\forall j \neq k : \mathbb{E}[\langle \boldsymbol{g}_j, \boldsymbol{g}_k \rangle] = 0$, measured by $\boldsymbol{m}_{t-1} \perp \boldsymbol{g}_t$ does not strictly hold in this case. In fact it seems that consequent updates point slightly in opposite directions. This means that we expect additional negative terms in Equation (17) and thus to over-estimate the approximated RMS update size $\eta_g$ in Equation (18).

Even though, we notice that the random walk model is only an approximation, our predictions hold fairly well, as evident from the last row.

## H    ROTATIONAL DYNAMICS OF SCALE-SENSITIVE PARAMETERS

Most neural network architectures have some scale-sensitive parameters. This commonly includes gains and biases as well as a final fully connected layer that is typically not followed by normalization. In networks without normalization, with infrequent normalization, or poorly placed normalization, most weight vectors can be scale-sensitive. The original, un-normalized, VGG (Simonyan and Zisserman, 2015) architecture is a good example of this, it consists of a series of convolutional layers with ReLUs and occasional pooling layers between them and series of fully connected layers towards the end. In this section we use it to investigate the rotational dynamics of scale-sensitive weights.

First we would like to note that the magnitude of scale-sensitive weights can also be largely arbitrary. Although they can't be scaled directly without affecting the loss, we can often scale two of them without affecting the network output. Consider two successive layers with a ReLU between them:

$$f(\boldsymbol{X}, \boldsymbol{W}_1, \boldsymbol{W}_2, \boldsymbol{b}_1, \boldsymbol{b}_2) = \text{ReLU}(\boldsymbol{X}\boldsymbol{W}_1 + \boldsymbol{b}_1)\boldsymbol{W}_2 + \boldsymbol{b}_2 \tag{96}$$

where $\boldsymbol{W}_1, \boldsymbol{W}_2 \in \mathbb{R}^{C \times C}$ are weight matrices, $\boldsymbol{b}_1, \boldsymbol{b}_2 \in \mathbb{R}^{1 \times C}$ are vectors, $\boldsymbol{X} \in \mathbb{R}^{B \times C}$ are inputs and we broadcast the operations. Note that the ReLU is positively homogeneous, so for a positive scalar $r > 0$ we have:

$$f(\boldsymbol{X}, r\boldsymbol{W}_1, r^{-1}\boldsymbol{W}_2, r\boldsymbol{b}_1, \boldsymbol{b}_2) = \text{ReLU}(\boldsymbol{X}\boldsymbol{W}_1 r + \boldsymbol{b}_1 r)\boldsymbol{W}_2 r^{-1} + \boldsymbol{b}_2 = f(\boldsymbol{X}, \boldsymbol{W}_1, \boldsymbol{W}_2, \boldsymbol{b}_1, \boldsymbol{b}_2) \tag{97}$$

Assuming the weights are scaled in-place (i.e. we don't modify the computation graph, only the weight values), this type of rescaling operation scales the relative update of $\boldsymbol{W}_1$ by $r^{-2}$ and $\boldsymbol{W}_2$ by $r^2$ when optimizing using SGD. This can significantly affect the learning dynamics as studied in e.g. Path-SGD (Neyshabur et al., 2015).

For a scale-sensitive weight $\boldsymbol{\omega}$, the gradient orthogonality (1) and inverse scaling (2) do not necessarily hold. The inverse scaling holds in terms of rescaling operations like the ones mentioned above if they are applicable. Generally, the gradient has some radial component in the direction of the weight. The expected magnitude of this component depends on the average angle between the gradient and the weight as well as the expected gradient magnitude itself. If we separate the gradient into radial and perpendicular components and view the radial component as a modification of the weights decay, we

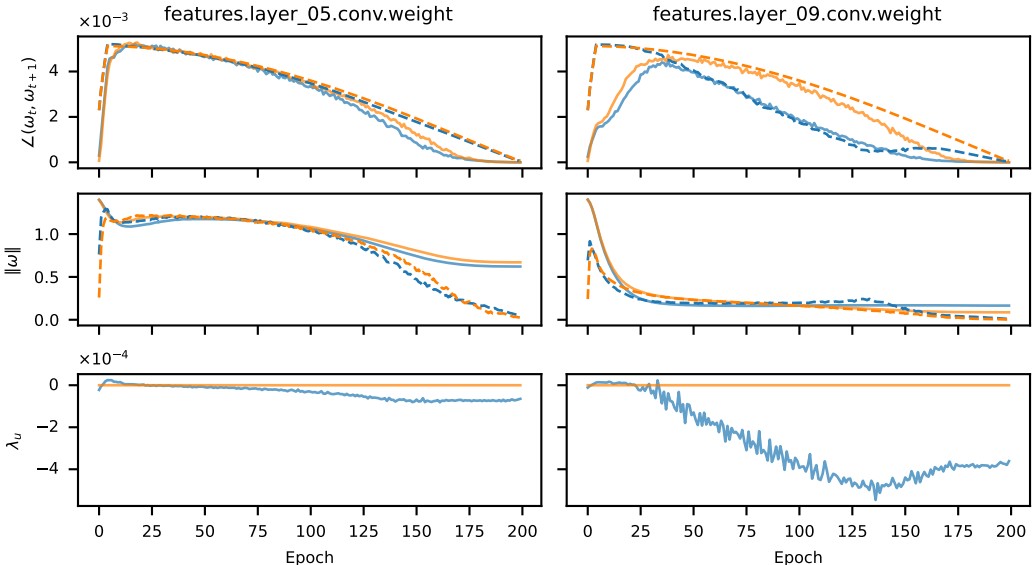

Figure 12: Measured (solid) and predicted equilibrium values (dashed) when training unnormalized (blue) and weight standardized (orange) variants of VGG-13 on CIFAR-10. The blue predictions account for the modified "effective" weight decay caused by the radial component of the gradient.

have a very similar setup to the one we analyzed for scale-invariant weights. If a stable equilibrium exists, this could give rise to rotational dynamics which may vary from weight to weight based on the "effective weight decay" for each one.

We explore this with VGG-13 training on CIFAR-10 using SGDM. We compare two versions, a standard unnormalized one and a variant where weight standardization is applied to every convolutional and fully connected layer. For each one, we measure the angular updates, the weight norms and the relative radial gradient magnitude:

$$\lambda_u = \mathbb{E}[\langle \boldsymbol{\omega}, \nabla_{\boldsymbol{\omega}}\mathscr{L}\rangle / \|\boldsymbol{\omega}\|^2] = E[\cos\left(\angle\left(\boldsymbol{\omega}, \nabla_{\boldsymbol{\omega}}\mathscr{L}\right)\right) \cdot \|\nabla_{\boldsymbol{\omega}}\mathscr{L}\| / \|\boldsymbol{\omega}\|] \qquad (98)$$

Note that we have written this in the case of no momentum by using $-\eta\nabla_{\boldsymbol{\omega}}\mathscr{L}$ instead of $\boldsymbol{u}$, but for the standard implementation of SGDM the momentum magnifies both this version of $\lambda_u$ and the standard "weight decay" ($L_2$ regularization) term the same way so they are comparable. The $\lambda_u$ term can therefore be viewed as modifying the weight decay, the effective weight decay parameter is $\lambda_e = \lambda + \lambda_u$ and accounts for the entire radial portion of a weight update. We replace the standard $\lambda$ with $\lambda_e$ when showing predicted values for the unnormalized network.

The results can be seen in Figure 12 for two weights. For the first one on the left, $\lambda_u$ is relatively small compared to $\lambda = 5 \cdot 10^{-4}$ and the weight behaves similarly in both setups, showing "standard" Spherical Motion Dynamics in the unnormalized setup. The equilibrium predictions match well early in training after the initial transition phase but the weight falls out of equilibrium towards the end when it can't decay fast enough to keep up with the equilibrium weight magnitude. For the second weight shown on the right, $\lambda_u$ is large causing a significant difference between the scale-invariant and scale-sensitive setups. The modified equilibrium predictions using $\lambda_e$ capture the behavior well in the middle phase of training, after the initial transition before the weight falls out of equilibrium towards the end. We note that in the unnormalized setup $\lambda_u$ changes over the course of training, starting out around 0 corresponding to an orthogonal gradient and growing larger in the later phases. This is likely due to the cross-entropy loss used, which is minimized with large (infinite) output magnitudes once the network has learned to accurately classify (overfit) the training data.

Our results for VGG13 suggest that scale-sensitive weights can also have rotational dynamics in real neural networks. The dynamics are less regular than in the normalized setup, with weights rotating at different speeds depending on the size of the radial gradient component. The weight magnitude can also not vary freely like for scale-invariant weights, where we can trade off the weight decay and learning rate without affecting the dynamics much (once equilibrium is achieved). Using large

amounts of weight decay in unnormalized networks can bring the weight norms out of balance, resulting in issues like vanishing gradients or activations. In unnormalized networks the magnitude of one weight matrix also affects the gradient magnitude of all others layers, further complicating the effect of weight decay. Our rotational optimizer variants constrain the dynamics to match the equilibrium dynamics of weight standardized networks throughout training, eliminating some of these effects.

## I   ROTATIONAL OPTIMIZER WRAPPER

In this section, we provide further details on the algorithmic design choices used in our rotation optimizer wrapper, as shown in Algorithm 1. Note that the method can act as a wrapper around any given existing optimizer F with a known $\widehat{\eta}_r$. In cases where the true value is unknown or undesirable, we can also specify some different value of our choice.

**Rotational and Non-Rotational Updates:** We use $\Omega$ to specify weights we apply rotational updates to, so a parameter $\boldsymbol{p}$ is treated differently based on whether $\boldsymbol{p} \in \Omega$ or not. Note that we can choose the scale of the weights in $\Omega$ as well, e.g. each filter can be considered on its own or as a part of a larger group such as the whole layer. The rotational wrapper leaves the update of non-rotational parameters unchanged. Rotational parameters are rotated by $\widehat{\eta}_r$ on average and their magnitude is kept constant. In practice, we may choose to treat some scale-sensitive weights as rotational, constraining their magnitude. Since their magnitude can actually matter for efficient learning, we optionally introduce a learnable gain to allow the network to learn the right magnitude for these weights. This gain can be absorbed into the weights for inference.

**Keeping the weight magnitude constant:** Alternatively, we could vary the weight magnitude according to our derived value for the equilibrium norm. However, with a learning rate schedule this value can become arbitrarily small causing numerical issues. For scale-invariant weights the magnitude doesn't matter so we simply keep it constant. This has the added benefit of removing the inverse scaling effect of the weight norm on the gradient magnitude (2), potentially making it a more meaningful metric.

**Controlling the rotation instead of the relative update:** The rotation of a scale-invariant weight $\boldsymbol{\omega}$ is generally caused by both $\Delta_g \boldsymbol{\omega}$ and $\Delta_\lambda \boldsymbol{\omega}$ as can be seen in Figure 2 and 3. In equilibrium, the sum of these components is roughly orthogonal to the weight vector. We want to avoid having to apply the weight decay and our constrained magnitude is generally not equal to the equilibrium magnitude. We therefore project $\Delta_g \boldsymbol{p}$ to be orthogonal to $\boldsymbol{p}$ and control the average size of this projected version of $\Delta_g \boldsymbol{p}$ instead of the original $\boldsymbol{p}$. This lets us explicitly control the angular update, regardless of any radial component in $\Delta_g \boldsymbol{p}$ that the weight decay would eliminate on average. If we apply rotational updates to scale-sensitive weights, performing Line 11 after Line 10 prevents any radial component in the gradient from affecting the rotational speed.

**Centering the weights:** Different normalization setups can result in slightly different SMD properties. Layer Normalization typically makes an entire weight matrix scale-invariant whereas Batch Normalization makes individual filters (i.e., rows or columns) independent. The default form of the rotational wrapper corresponds to the rotational equilibrium dynamics obtained with Weight Standardization (Qiao et al., 2019) also known as Centered Weight Normalization (Huang et al., 2017), where each filter is scale and shift invariant. We remove the mean $\bar{\boldsymbol{p}} = \frac{1}{C} \sum_{i=1}^{C} p_i$ of $\boldsymbol{p} = [p_1, \ldots, p_C]$ since it is irrelevant in this setup. This removal was also found to be beneficial in NF-Nets (Brock et al., 2021a;b).

**Hyperparameters:** The algorithm requires an $\varepsilon$ value for numerical stability but otherwise only adds one hyperparameter, a decay factor $\beta$ similar to those in Adam. It determines the rate at which we update our estimate of the average update magnitude (Line 11). This in turn controls how much we let the rotation vary between steps. We could potentially derive an analytical value for $\beta$ based on the convergence speed towards equilibrium. For example $\beta$ should perhaps be roughly equal to $\sqrt{a}$ from Equation (14) for AdamW, when trying to match the dynamics exactly. However, this rate may not be optimal and generally depends on the learning rate (which may be scheduled). We use a default of $\beta = 0.99$ which should keep the expected angular update close to the equilibrium value over time, while still allowing some variation from step to step. There is likely batch size dependence in the optimal value of $\beta$, with larger batches potentially benefiting from smaller values since balancing the

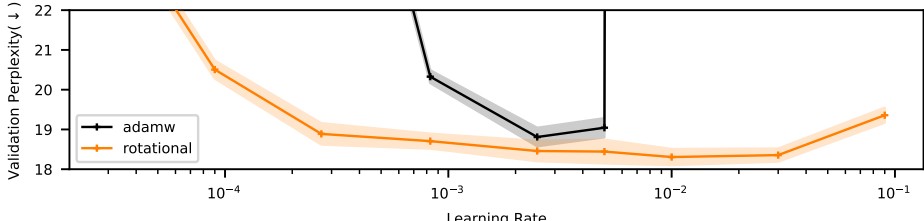

Figure 13: Validation perplexity for GPT-like model on Wikitext for different learning rate, weight decay pairs with a constant product ($\eta\lambda = 2.5\cdot10^{-3}$) resulting in a specific $\widehat{\eta}_r$ (Table 1).

Table 3: Test set performance for baseline optimizers SGDM and Lion and their RVs. Zero shot results for RVs use the baseline hyparparameters, the best shot is lightly tuned.

| Dataset | Model | Optimizer | Batch Size | Metric ($\updownarrow$) | Baseline | Zero Shot | Best Shot |
|---------|-------|-----------|------------|-------------------------|----------|-----------|-----------|
| CIFAR-10 | ResNet-20 | SGD | 128 | Top-1 Acc. ($\uparrow$) | 92.7 $\pm$0.10 | 92.6 $\pm$0.18 | N/A |
| CIFAR-10 | ResNet-20 | SGD | 2048 | Top-1 Acc. ($\uparrow$) | 92.0 $\pm$0.14 | 91.9 $\pm$0.11 | 92.1 $\pm$0.42 |
| CIFAR-10 | ResNet-20 | Lion | 128 | Top-1 Acc. ($\uparrow$) | 92.1 $\pm$0.12 | 91.7 $\pm$0.13 | N/A |
| CIFAR-10 | ResNet-20 | Lion | 2048 | Top-1 Acc. ($\uparrow$) | 91.8 $\pm$0.22 | 91.5 $\pm$0.12 | 91.6 $\pm$0.30 |
| Imagenet-1k | ResNet-50 | SGD | 256 | Top-1 Acc. ($\uparrow$) | 77.4 | 77.3 | N/A |

average rotation within in each step could be sufficient in these cases. An Adam-like bias correction is applied to the average update magnitude when it is used (Line 12).

**Resource Requirements:** We need to keep track of two scalars $\nu_{\boldsymbol{p}}$ and $n_{\boldsymbol{p}}$ for each rotational parameter. Since $\boldsymbol{p}$ is generally a vector, such as a row in a weight matrix, the memory requirement is negligible compared to quantities like momentum that store a scalar for every element of $\boldsymbol{p}$. The computational requirements in terms of floating-point operations are also relatively small, linear in the number of network (scalar) parameters like standard optimizers. However, the rotational variants are not applied fully elementwise, making efficient (parallel) implementations slightly harder.

## J   ADDITIONAL EXPERIMENTS

**Learning Rate vs Weight Decay for a Transformer model:** In this section, we replicate the experiment previously described using a GPT-like model trained on Wikitext. The outcomes are illustrated in Figure 13. Unlike the ResNet-20 model trained on CIFAR-10, for transformers, linear layers are not fully scale-invariant; the weight norms matters. Thus, as mentioned before, we introduce a learnable gain for these layers when training with our RVs. Varying the learning rate affects the updates to biases and gains in the RV. Thus we expect the performance of this network to be more sensitive to changes in the learning rate. Again, we believe the noticeable performance difference between AdamW and RV is primarily attributed to differences in their effective step size schedules.

**Constraining the Rotational Dynamics:** In this section, we examine the performance of our RV of SGDM and Lion. Table 3 confirms the results we have seen with AdamW for both SGDM and Lion. Without any additional hyperparameter tuning (zero-shot) or light tuning (best-shot), our RVs closely match the performance of the original variants. This supports the idea that we can simplify deep neural network (DNN) training by standardizing the update size to $\eta_r$ and skipping the initial phase.

**Best-Shot Sweep:** In practice, we have observed, that the difference between the initial learning rate, when the baseline optimizer has not yet transitioned to equilibrium, can provide a good starting point for best shot hyper-parameter tuning. Figure 14 depicts the tracked angular updates for the first 40 epochs of the DeiT tiny experiment on Imagenet-1k, as reported in Table 4. We can see that at beginning of training the learning rate of the baseline is approximately 2 times higher than for the updates of the RV. Using Table 2 and 4, we can confirm that a 2 times higher effective relative update allows the RV to match performance of the original optimizer.

**Hyperparameter Sensitivity:** The RVs introduce one hyperparameter, i.e., the decay rate $\beta$. Further, we can decide whether to enable (y) or disable (n) centering of the weights (zero-mean) and whether

to enable scale-invariance on tensor (t) or filter (c) level. In this section we study the sensitivity of these choices in two different setups. We train a ResNet-18 on a random train split of CIFAR-10 with the RV of SGDM and a GPT-like model on Wikitext with the RV of AdamW. The results are shown in Figure 15. They indicate that the performance of the RVs remains relatively stable when the hyperparameters are varied. In our Adam experiments, using a special RV that combines the update direction $\Delta_g \boldsymbol{p}$ from Adam+$L_2$ with the $\widehat{\eta_r}$ of AdamW, we have observed that varying $\beta$ noticeably affects the effective step size schedule and thus required slight tuning to increase performance.

## K  EXPERIMENTAL DETAILS

We perform our experiments on several popular datasets, i.e., CIFAR-10/100 (Krizhevsky, 2009) and Imagenet-1k (Russakovsky et al., 2015) for image classification, IWSLT2014 (Cettolo et al., 2014) for German-English translation, and Wikitext (Merity et al., 2017) for language modelling. Our code utilizes the TIMM library (Wightman, 2019) for vision tasks and FairSeq (Ott et al., 2019) for translation.

We train commonly used architectures, namely ResNet-20, ResNet-18, ResNet-50 (He et al., 2016), DeiT tiny (Touvron et al., 2021), a small transformer, and a small GPT-like network (Radford et al., 2019) from scratch.

**General Setup:** For all experiments trained with SGDM we use a momentum of 0.9, for experiments trained with AdamW we used $\beta_1 = 0.9$ and $\beta_2 = 0.999$ and for experiments trained with our RVs we used $\beta = 0.99$, unless otherwise stated. For Lion we used $\beta_1 = 0.9$ and $\beta_2 = 0.999$ exclusively. The experiments on Imagenet-1k have been run on a single V100-SXM2-32GB GPU. All other experiments are run on a single NVIDIA A100-SXM4-40GB GPU.

Note that we used default architectures for the baseline experiments, but used a learnable gain for DeiT trained on Imagenet-1k, Transformer-S trained on IWSLT2014 de-en and the GPT-like architecture on Wikitext. As mentioned in Section 4 this was necessary because for transformers the layers are not fully scale-invariant and the norm of the weights matters in practice.

Here we list additional details referenced in the **details**-column in Table 4 and 5:

D-1 We pre-process the data by normalizing it with mean $(0.4914\,0.4822\,0.4465)$ and std $(0.2023\,0.1994\,0.2010)$. For training we used simple data augmentation from He et al. (2016).

D-2 We use the standard data augmentation from He et al. (2016) for Imagenet.

D-3 Analogously to Touvron et al. (2021) we apply strong data augmentation. We use color jitter brightness up to 0.3, auto-augmentation, random erase with probability 0.25, drop path with probability 0.1, mixup with probability 0.8 and cutmix with probability 1.0. Additionally, we use label smoothing of 0.1.

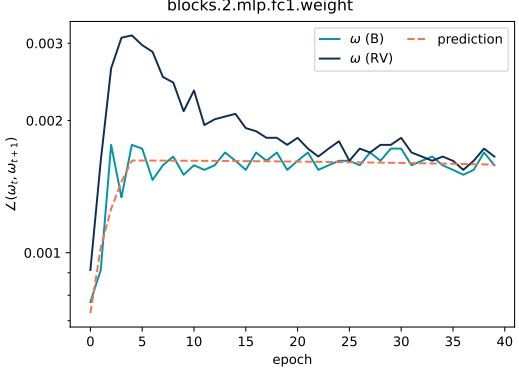

Figure 14: $\eta_r$ measured for the first 40 epochs of a DeiT tiny trained on Imagenet-1k. Before AdamW transitions to equilibrium we observe a difference of roughly two times higher learning rate compared to its rotational variant.

| Dataset | Model | Zero Mean | Invariance |
|---|---|---|---|
| CIFAR-10 | ResNet-18 | 94.6±0.19 (**yc**) 94.9±0.10 (**nc**) | 94.6±0.19 (**yc**) 94.6±0.47 (**yt**) |
| Wikitext | GPT-like | 18.5 ±0.42 (**yc**) 18.5 ± 0.42 (**nc**) | 18.5±0.42 (**yc**) 18.4±0.15 (**yt**) |

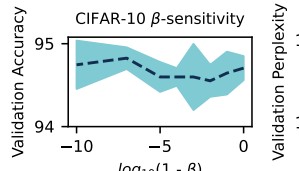
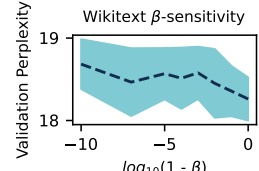

Figure 15: Experimental results for hyperparameter sensitivity. We report perplexity (↓) on Wikitext validation dataset and top-1 Acc. (↑) on a random validation split on CIFAR-10. In the default set up weight centering (y) and per filter (c) scale-invariance is enabled.

Table 4: Experimental set up (include training set and test set definition).

| Dataset | Model | Optimizer | Batch Size | zero shot | best shot | Wrapped Adam+$L2$ | details | lr schedule | warmup | epochs (e)/ iteration (it) | train duration | precision |
|---|---|---|---|---|---|---|---|---|---|---|---|---|
| CIFAR-10 | ResNet-20 | SGD | 128 | wd=1e−4 lr=0.5 | N/A | N/A | (D-1) | cosine | lr=1e−6 5 epochs | 200 (e) | 35min | float32 |
| CIFAR-10 | ResNet-20 | SGD | 2048 | wd=1e−4 lr=4.8 | wd=1e−4 lr=16 | N/A | (D-1) | cosine | lr=1e−6 5 epochs | 200 (e) | 35min | float32 |
| CIFAR-10 | ResNet-20 | AdamW | 128 | wd=1e−2 lr=5e−2 | N/A | $\beta = 0.9$ | (D-1) | cosine | lr=1e−6 5 epochs | 200 (e) | 35min | float32 |
| CIFAR-10 | ResNet-20 | AdamW | 2048 | wd=1e−2 lr=1.6e−1 | wd=1e−2 lr=0.8 | $\beta = 0.0$ | (D-1) | cosine | lr=1e−6 5 epochs | 200 (e) | 35min | float32 |
| CIFAR-10 | ResNet-20 | Lion | 128 | wd=1.0 lr=5e−4 | N/A | N/A | (D-1) | cosine | lr=1e−6 5 epochs | 200 (e) | 35min | float32 |
| CIFAR-10 | ResNet-20 | Lion | 2048 | wd=1.0 lr=1.6e−2 | wd=1.0 lr=8e−2 | N/A | (D-1) | cosine | lr=1e−6 5 epochs | 200 (e) | 35min | float32 |
| Imagenet-1k | ResNet-50 | SGD | 256 | wd=1e−4 lr=1e−1 | N/A | N/A | (D-2) | cosine | lr=1e−6 5 epochs | 90 (e) | 30h | float16 |
| Imagenet-1k | DeiT tiny | AdamW | 1024 | wd=5e−2 lr=5e−4 | wd=1e−1 lr=5e−4 | N/A | (D-3) | cosine | lr=1e−6 5 epochs | 300 (e) | 70h | float16 |
| CIFAR-10 | DeiT tiny | AdamW | 64 | wd=5e−2 lr=5e−4 | N/A | $\beta = 0.99$ | (D-3) | cosine | lr=1e−6 5 epochs | 600 (e) | 16h | float16 |
| IWSLT2014 de-en | Transformer-S | AdamW | 4096 | wd=1e−4 lr=5e−4 $\beta_2 = 0.98$ | wd=2e−1 lr=5e−4 $\beta_2 = 0.98$ | $\beta = 0.99$ | (D-4) | cosine | 4000 (it) | 22021 (it) | 50min | float16 |
| Wikitext | GPT-like | AdamW | 55 | wd=0.5 lr=5e−3 $\beta_2 = 0.95$ | N/A | $\beta = 0.9$ | (D-5) | cosine div_f=1e2 final_div_f=1e4 | 2e−2(%) | 15000 (it) | 3h | bfloat16 |

Note that we used image up-scaling when DeiT tiny is trained on CIFAR-10 because DeiT tiny is designed for Imagenet images.

D-4 We use standard FairSeq library (Ott et al., 2019) with dropout probability 0.3.

Note that additionally to using weight standardization with learnable gain, we set the weight decay to 0 for scale-sensitive weights. This is done by default for the vision tasks in TIMM library (Wightman, 2019), but not by FairSeq library (Ott et al., 2019) we used for this task. We observed no difference in performance for the baseline model, yet this adjustment allowed to tune the effective learning rate for the scale-invariant weights, without affecting the learning rate of the scale-sensitive weights significantly.

D-5 For this experiment we use the llm-baseline library library (Pagliardini, 2023). For the GPT-like architecture, we use vocabulary size of 50304, sequence length of 512, embedding size of 768. The model features 12 repeated blocks, each comprising a self-attention block followed by a two-layer MLP block with hidden dimension 3072. This results in a total of 124 million parameters. For the drop out layers we use a probability of 0.2.

**Constraining the Rotational Dynamics:** The experimental details for the experiments reported in Table 2 and 3 can be found in Table 4. Note that for the Wrapped Adam+$L_2$ experiments we used the best shot settings. For these experiments, we have observed that varying $\beta$ noticeably affects the effective step size schedule. Thus we used minimal hyperparameter tuning for $\beta$. The parameter configuration is shown in the respective column of Table 4.

**Learning Rate vs Weight Decay:** For the ResNet-20 experiment on CIFAR-10 and language model task on Wikitext with a GPT-like model we use the best shot (zero shot) setting reported in Table 4 as

Table 5: Experimental details for hyperparameter senstivity study.

| Dataset | Model | Optimizer | training dataset | validation dataset | hyper-parameters | details | lr schedule | warm up | epochs (e)/ iterations (it) | train duration | precision |
|---------|-------|-----------|------------------|--------------------|-----------------|---------|-------------|---------|------------------------------|----------------|-----------|
| CIFAR-10 | ResNet-18 | SGD | 90% Train | 10% Train | wd=1e−4 lr=0.5 | (D-1) | cosine | lr=1e−6 5 epochs | 200 (e) | 35min | float32 |
| Wikitext | GPT-like | AdamW | Train | Validation | wd=0.5 lr=4e−3 $\beta_2 = 0.95$ | (D-5) | cosine div_f=1e2 final_div_f=1e4 | 2e−2 (%) | 15000 (it) | 3h | bfloat16 |

default. We then sweep over the learning rate keeping $\eta\lambda = 5\cdot10^{-4}$, $\eta\lambda = 2.5\cdot10^{-3}$ respectively, constant.

**Scheduling Effects:** In the experiment described in the preceding paragraph, we monitored $\|\boldsymbol{\omega}\|$, $\angle(\boldsymbol{\omega}_t, \boldsymbol{\omega}_{t+1})$ during training for one of the runs with the chosen learning rates: $1\cdot10^{-4}$, $8.3\cdot10^{-3}$, and $3\cdot10^{-1}$.

**Adam vs AdamW:** For the sweep we train a ResNet-18 on a 90/10 train/val split from the original train set. We use a step-wise cosine schedule and train for 200 epochs without warmup. In this sweep we try to reproduce the results in Figure 2 from Loshchilov and Hutter (2019), albeit with a slightly different network and training for 200 epochs instead of 100. The best configuration for Adam was $\eta = 7.813 \cdot 10^{-4}$, $\lambda = 1.250 \cdot 10^{-4}$ resulting in a validation set accuracy of 93.919. The best configuration for AdamW was $\eta = 1.25 \cdot 10^{-2}$, $\lambda = 8.0 \cdot 10^{-2}$ with a validation accuracy of 94.319. On the test set we run each configuration over 5 different seeds, using the AdamW hyperparameters for both RV-AdamW and the special wrapped Adam+$L_2$ RV. The results were Adam+$L_2$ $94.08 \pm 0.16$, AdamW $94.74 \pm 0.14$, RV-AdamW $94.57 \pm 0.12$ and the special wrapped Adam+$L_2$ $94.55 \pm 0.07$. Note that the performance of the two RVs is almost identical and higher than standard Adam+$L_2$, but slightly lower than AdamW which is not surprising for a zero-shot transfer of a well tuned baseline and likely due to scheduling effects.

**Imbalanced Rotation:** We trained a ResNet-18 on a 90/10 train/val split from the original train set with weight decay $\lambda = 0.01$ and varying learning rates $\eta \in \{2.7\cdot10^{-3}, 8.3\cdot10^{-3}, 2.5\cdot10^{-2}, 5\cdot10^{-2}, 1\cdot10^{-1}, 3\cdot10^{-1}, 0.9\cdot10^{-1}, 4.5\}$). For each rotation speed scaling $f$ and portion $p$ we report the best performance of this sweep. All other settings are equivalent to the settings reported in Table 4.

**Training Poorly Normalized Networks:** We train a ResNet-18 with layer normalization on CIFAR-100 using the same augmentation, learning rate schedule and base hyperparameters as for the ResNet-20 on CIFAR-10 experiments, unless otherwise noted below. We train on a random subset containing 90% of the train set and use the remaining 10% for validation which we report. The inputs are normalized for mean (0.5071, 0.4867, 0.4408) and std (0.2675, 0.2565, 0.2761). We use a weight decay of $5 \cdot 10^{-4}$ and a batch size of 256. The layer normalization is implemented with a Group Normalization (Wu and He, 2018) using a single group.

**Need for Learning Rate Warmup:** These experiments follow the same base setup as we report in Table 4. We train for a total of 10 epochs using a cosine decay schedule (applied stepwise) and no warmup. We use local accumulation on top of batches of size 256 to emulate larger batch sizes.

**Hyperparameter Sensitivity:** The experimental details for the experiments reported in Figure 15 can be found in Table 5.

