# OpenReview forum: "Rotational Equilibrium: How Weight Decay Balances Learning Across Neural Networks"
_ICLR.cc/2024/Conference — Submitted to ICLR 2024_

### Official Review · Reviewer_Fts9 · 2023-10-29

**Soundness:** 3 good
**Presentation:** 3 good
**Contribution:** 3 good
**Rating:** 5
**Confidence:** 3

**Summary:**

This work explores how the combined effect of weight decay and gradient update drives GD iterates to a rotational equillibrium for various optimizers. The authors propose that this rotational equiliibrium can be achieved early in training and how achieving this equillibirum can play a key role in effectiveness of AdamW compared to Adam with L2-regularization. To study the equillibirum point, the authors analyze a simplified setting, the optimization of a random loss function, resulting in a random walk for the neural network parameters and derive this point for various optmizers.

**Strengths:**

1) The paper is easy to read and follow.
2) This work attempts to understand an important problem.

**Weaknesses:**

Here are some of my major concerns:
1) I doubt comparing the dynamics of random walk (with zero mean gradients) with neural network training with true objective is meaningful. In particular, it is not clear how a random walk (drawing gradients from a normal 0 mean distribution) can trace the dynamic of neural network trained with a true loss.
2) The authors state *"Although the noise component can easily depend on the the progress on the underlying objective function, we can
view the random walk as an approximation of this noise component."* but the random walk is not a function of the true loss, hence making it unclear how it is close to real training dynamics. It maybe possible that I misunderstood, so I will wait for further clarification from the authors.
3) "This causes the expected rotation of the vector in each update to remain constant along with its magnitude.": Although the GD iterates converge in some sense but for moderate lr GD, oscillates in the edge of stability regime https://arxiv.org/abs/2103.00065. Hence, it is not necessarily true that expected rotation in each update remains constant.

Minor:

4) In figure-2,3, the weight norm equillibrium is a scalar, but denoted as a vector in the figure, does the author mean the weight vector at equillibrium?
5) Similarly for figure-3, please redefine the figure as the expected quantities are scalars but shown as a vector.
6) Define what is the expectation over, when defining the angular update?

**Questions:**

1) In particular, the justification of neural network random walk with actual training is not clear. I would like to hear from the authors regarding this.
2) Additionally, it is important than authors correct the terminologies.

---

> ### Author Response · Authors · 2023-11-14
>
> Thank you for the time spent reviewing our work as well as your feedback and suggestions.
> Below we will try to address the weaknesses you bring up and answer your questions.
>
> ## Random Walk:
> First, we would like to provide further explanation of our random walk setting and why we consider it relevant to real neural networks. We apologize for not making this clearer in the manuscript and will update it to include the following discussion.
>
> ### Relation between random walk and real training dynamics:
> Let's assume we are in the standard empirical risk minimization setting used in deep learning.
> We have a training dataset $\mathbb{X}$ consisting of points $\\{\mathbf{x}\_{1} \ldots \mathbf{x}\_{N}\\}$, where each point accounts for both an input and a label/target when applicable.
> Let $\mathbf{\theta}$ be the whole parameter vector of our model, accounting for all layers etc.
> We can then write our full training loss as:
> $$ \mathscr{L}(\mathbb{X}, \mathbf{\theta}) = \frac{1}{N}\sum\_{i=0}^{N} \mathscr{L}(\mathbf{x}\_i, \mathbf{\theta})$$
> Similarly we can write the loss for a minibatch $\mathbb{B}$ of size $|\mathbb{B}|$ as:
> $$ \mathscr{L}(\mathbb{B}, \mathbf{\theta}) = \frac{1}{|\mathbb{B}|}\sum\_{\mathbf{x}\_i \in \mathbb{B}} \mathscr{L}(\mathbf{x}\_i, \mathbf{\theta})$$
> The full-batch gradient is $\mathbf{g}\_\text{True} = \nabla\_{\mathbf{\theta}}\mathscr{L}(\mathbb{X}, \mathbf{\theta})$ and the mini-batch gradient is $\mathbf{g}\_\mathbb{B}=\nabla\_{\mathbf{\theta}}\mathscr{L}(\mathbb{B}, \mathbf{\theta})$.
> The noise component of the gradient, $\mathbf{g}\_N$ is then the difference between the two:
> $$\mathbf{g}\_N = \mathbf{g}\_\mathbb{B} - \mathbf{g}\_\text{True}$$
> Note that this noise component is zero-mean, since the minibatch gradient is an unbiased estimate of the full batch gradient for a randomly sampled $\mathbb{B}$:
> $$\mathbb{E}_{\mathbb{B}}[\mathbf{g}\_\mathbb{B}] = \mathbf{g}\_\text{True}$$
> Our random walk analysis assumes that the batch gradient is dominated by the noise component, i.e. that:
> $$\mathbf{g}\_\mathbb{B} = \mathbf{g}\_\text{True} + \mathbf{g}\_N \approx \mathbf{g}\_N$$
> This does of course not hold exactly in practice but may be a reasonable approximation, especially for smaller batch sizes.
>
> We note that the noise component is always zero-mean in expectation which is the main requirement for our random-walk analysis.
> The presence of normalization also helps ensure that the gradient is orthogonal in expectation even when $\mathbf{g}_\text{True}$ is significant.
>
> ### Relevance of Random Walk Setting:
> We chose this random walk setting for several key reasons:
> 1) It is a single relatively simple assumption that streamlines the math of the analysis.
> Without it we would have to make various other approximations or assumptions about the gradients or alternatively introduce unknown quantities describing aspects like the alignment of gradients over time.
> This would result in a more complicated analysis that is harder to understand as well as equilibrium predictions that may depend on unknown quantities.
> 2) It allows us to extend the analysis to the unnormalized setting, unlike prior work which has only focused on the fully scale-invariant case.
> This allows us to provide a broader view of the effects of weight decay in deep learning since most networks are not perfectly scale-invariant in practice.
> 3) It captures important properties of the gradient unlike simpler models that might for example sample the weight gradient directly from a fixed distribution.
> Our setup accounts for e.g. the presence of normalization layers so $\mathbf{g}_N$ respects fundamental properties like the orthogonality and the inverse proportionality (Equations 1 and 2 in the paper).
> For a real optimization problem the distribution of $\mathbf{g}_N$ may change with the progress on $\mathscr{L}(\mathbb{X}, \mathbf{\theta})$, but we believe that if this happens slowly enough compared to the convergence towards equilibrium it won't significantly affect the results (and this would likely be very hard to analyze anyway).
>
> **Finally, it is important to highlight that the predictions we derive from analyzing this random walk scenario closely match our measurements of real neural networks in practice.**
> In Appendix G, we explore different scenarios and provide examples illustrating the accuracy of these predictions, showing a close alignment, often within a small percentage margin.
> **We therefore believe our analysis offers valuable insights and predictions applicable to real neural networks, despite being derived for a simplified setting.**

---

> > ### Author Response · Authors · 2023-11-14
> >
> > ## Relation to Edge of Stability:
> > We note that our analysis focuses on noisy SGD whereas Edge of Stability focuses on full batch gradient descent.
> > It is also unclear whether they use weight decay in their experiments, another key component of our analysis.
> > We do therefore not believe our findings are contradicted by theirs.
> > It is quite likely that a proper equilibrium can not form in a fully noiseless setting where the gradient distribution can change drastically between steps.
> >
> > ## Vector vs Scalar Quantities in Figures
> > Apologies, we do not fully understand what you mean here.
> > Is the issue that we depict vectors (with the arrowheads) but denote their magnitude in the figures?
> >
> > ## Expectation
> > Thank you for bringing this to our attention.
> > The random walk is an ergodic process so the expectation can be seen as either being across all possible random trajectories (i.e. expectation over random initial seeds) or across all timesteps along a single trajectory (i.e. expectation over time for a given seed).
> > We note that it does not hold for a given timestep when conditioned on previous ones.
> > For example, after a large step, the weight magnitude is likely to exceed the equilibrium value, leading to smaller expected rotations until the weight magnitude returns to the equilibrium range.
> >
> > ## General Remark
> > We will incorporate your feedback clarifying the points you brought up.
> > Please let us know if you have further questions.
> > If you feel our response adequately addresses your concerns, we would greatly appreciate it if you would consider raising your review score.

---

> ### Author Response · Authors · 2023-11-21
>
> We hope this message finds you well. We understand your time is precious and deeply appreciate your efforts in this review process. Since the discussion period is ending soon, we would like to kindly ask for your feedback on our rebuttal. If you feel it addresses at least some of your concerns, we would greatly appreciate it if you updated your score accordingly.
>
> Please let us know if you have any further questions and we will quickly get back to you.

---

> > ### Comment · Reviewer_Fts9 · 2023-11-21
> > **Reviewer response-1**
> >
> > I thank the authors for their detailed feedback and changes made in the manuscript, it would have been easier for the reviewers to read the changes if it was highlighted with a different color say red.
> >
> > My main criticism from the random walk approach in particular was the simple question: if we use $g_{N}$ doesn't it lose the information about the original loss it was supposed to minimize $L(X,\theta)$. How much of the information of the original loss is conveyed in this $g_{N}$? Apparently it does not seem clear how the information in the loss can be conveyed through this noisy gradient $g_{N}$.
> >
> > Regarding scalar vs vector issue, now it is clear to me. Thanks.

---

> > > ### Author Response · Authors · 2023-11-21
> > >
> > > Thank you for getting back to us. We have uploaded a new revision where we have highlighted the changes in red that will hopefully make them easier to review (thank you for this suggestion).
> > >
> > > **Regarding your question about the random walk, in short: Yes, it will lose information about the loss it is supposed to minimize. However, we do not believe this affects the equilibrium dynamics significantly (at least in many cases of interest).**
> > >
> > > First we would like to emphasize that **equilibrium does not require the loss to converge, it is a separate process that happens entirely because of the orthogonal geometry of the updates (not their exact direction)**. If the batch gradient is dominated by the noise component the geometry of these updates will look similar to a random walk, at least at the start of training before we make significant progress on the objective. Also note that $\mathbf{g}_N$ respects all the fundamental properties of the gradient such as its dependency on the weight magnitude and orthogonality conditions from the use of normalization layers, the random walk respects these constraints imposed by the network architecture.
> > >
> > > When optimizing a real loss the distribution of $\mathbf{g}_N$ may change over time in a way that the random walk does not model, i.e. because of the progress on the underlying objective. This is something that is very hard to model (we would at least need to make additional assumptions about the form of the loss function and the dataset etc). However, **if this change in $\mathbf{g}_N$ is slow compared to how fast the system converges to equilibrium, we can approximate the gradient distribution with a fixed one over shorter time scales**. This is similar to our assumption of a fixed learning rate, if it changes slowly enough the system can roughly stay in equilibrium corresponding to the current lr value at any given time. Note that in many cases the predictions in Table 1 do not depend on the exact distribution of the gradient.
> > >
> > > **In practice we observe that the results of our analysis hold very well for the real neural network optimization problems we tested them on.** There will be some differences e.g. successive gradients may not be exactly orthogonal in expectation. This slightly affects the results like we discuss in Appendix G but the final predictions hold remarkably well for the problems we tested. We expect the differences might be greater for sufficiently large batch sizes where the noise component is not dominant. We use standard batch sizes in our experiments (not artificially small) and the predictions hold well.
> > >
> > > The random walk is of course only an approximation of the dynamics observed in real neural network optimization. However, **we believe it strikes a good balance between the accuracy of the predictions (very high) compared to the complexity of the analysis (quite reasonable)**. It also seems probable that modelling the exact dynamics more closely may simply introduce unknown quantities (that need to be measured) into the predictions in Table 1 which would not make them more useful in practice, if the closed form solutions exist at all.
> > >
> > > We hope this clarifies this point but please let us know if you have further questions.

---

### Official Review · Reviewer_tLyV · 2023-11-01

**Soundness:** 2 fair
**Presentation:** 1 poor
**Contribution:** 2 fair
**Rating:** 3
**Confidence:** 3

**Summary:**

This paper studies the interplay between the weight decay and the gradient updates. The underlying phenomenon being studied is related to equilibrium state, which is achieved when the gradient update and the weight decay cancel each other out. In such a state, the parameter updates keep the magnitude same but rotate the update. This work characterizes such equilibrium for various layers in a neural network, for different optimizers including AdamW and SGD. It also argues that rotational equilibrium is the root cause for the performance gap between AdamW and Adam+L2 regularization. Finally, this work also proposes routine to enforce rotational equilibrium in various optimizers.

**Strengths:**

- Empirical analysis to explain the difference between AdamW and Adam+L2 regularization using rotational equilibrium is promising
- Simple procedure to create rotational variants of the existing optimizers

**Weaknesses:**

- It is unclear what the advantage of the rotational variants of an optimizer like AdamW over the vanilla optimizer. Paper does not do any justifications as to why one would prefer rotational variants in practice over the vanilla counterparts given that there is a non-trivial change required. For instance, Table 2 does not convincingly says that the RV AdamW would be the preferred choice.
- Paper structure makes too many keypoints which hinders the impact of the really important aspects of this work. Since this work is more on the empirical analysis, it would be beneficial if fewer experiments are discussed in details, rather than discussing many experiments in few paras.
- Need clarifications on many observations made in the paper (see questions below)

**Questions:**

- In Figure 4, what do the colors correspond to? Do these correspond to different neurons or different layers?
- In Table 1, do you have any intuition as to why the RMS update for Lion is devoid of any beta terms? In contrast, AdamW updates depend on both beta and C.
- In Table 2, can you explain the performance degradation in RV-AdamW Zero Shot on dataset IWSLT2014 de-en? Also, why is the Wrapped Adam+L2 missing for the ImageNet-1k dataset?
- In Figure 8, you show the ImageNet-1k+ResNet50 performance for 10 epochs. Does the trend hold true for longer training? Do the baselines and RV variant achieve significantly different Top-1 accuracy?

---

> ### Author Response · Authors · 2023-11-14
>
> Thank you for taking the time to review our paper and your feedback. We will try to address your concerns and questions below.
>
> On a high level, **we believe there may have been some misunderstanding about the main goal of our paper which is understanding the effect of weight decay on the learning dynamics of existing optimizers, not proposing a new SOTA optimizer.**
> Aside from the high level structure of the manuscript, we have tried to further emphasize this in various places e.g. when proposing the RVs *"In this section we create an experimental tool to study these phenomena"* (referring to the effects of the transient phase and imbalanced rotation) and in the experiment section *"Although the RVs have many interesting properties and good performance, we view them primarily as a research tool"*.
>
> ## The Advantage of Rotational Optimizers
> The main purpose of the RVs in this work is to serve as research tools to further our understanding of weight decay and balanced rotation in deep learning.
>
> We model the RVs to match the equilibrium dynamics of the standard optimizers (with proper normalization) as closely as possible.
> **By design, this leaves no room for changes that could bring significant performance differences except in cases where the original optimizers deviate significantly from the regular "balanced" equilibrium dynamics.**
> When that is the case the rotational variants tend to perform better, see for example the poorly normalized case (imbalanced rotation) and the need for warmup (harmful transient phase) sections of our experiments.
>
> The main conclusions we draw from Table 2 is not that the RVs perform better but rather:
> * The transient phase is not needed, the simpler enforced equilibrium dynamics of the RVs suffice for good performance.
> * In many cases we can predict what the hyperparameters of the RVs should be (zero-shot transfer). This supports our analytical results since we can predict what the rotational update size should be.
>
> If one wants to use RVs as practical optimizers (not what we propose them for), we believe the main advantage would be simpler and more robust learning dynamics. For example:
> * Enforced balanced rotation even in cases with poor / no normalization (more robust to the neural network design).
> * Eliminating the transient phase, giving better learning dynamics early in training (likely faster progress, reduced need for warmup).
> * Allowing the direct specification of the "effective learning rate schedule" (in terms of the update sizes), otherwise it depends on interactions between the specified schedule with the initialization, weight decay and learning rate used.
>
> On their own these benefits might not be significant enough to warrant switching to the RVs for mainstream, well established / tuned training tasks.
> Popular architectures are likely to have at least somewhat decent dynamics through "natural selection", even if obtained through trial and error.
> However, these are desirable properties that may inspire future optimizer design (where there is freedom to deviate arbitrarily from existing optimizers, unlike with the RVs).
>
> ## Paper Structure
> We don't consider this work primarily empirical, all of our experiments aim to validate our analysis or quantify the impact of specific phenomena observed in our analysis such as the transient phase, imbalanced rotation, different roles of learning rate and weight decay etc.
>
> Our analysis reveals a broad range of insights that we empirically explore / validate in our experiments.
> We believe this structure makes sense for this purpose.
> If our main goal was to propose a new SOTA optimizer, we agree that this structure would not be optimal and a narrower focus on showing specific advantages would be better.
> We note that our discussion section further interprets and summarizes our experimental results.

---

> > ### Author Response · Authors · 2023-11-14
> >
> > ## More / Longer Experiments on ImageNet
> > This is something we would like to do but our compute resources are on the order of 1 GPU.
> > If we had 10s or 100s we would definitely include more large scale / compute intense experiments.
> > **Note again that the main focus of this work is understanding, not achieving a particular score for a specific experiment. In that sense we believe the shorter i1k runs are sufficient to show that the chaotic transient phase can be detrimental to training if a learning rate warmup is not used.**
> > This is not only relevant for obtaining a better performance with the RVs but also of wider interest, for example:
> > * It shows that the transient phase matters and is something practitioners should be aware of. Even if they don't use a relative / rotational optimizer, analyzing the transient phase could potentially inform them about how long the warmup should be.
> > * This could explain why existing relative optimizers like Lars and Lamb seem to have a reduced need for a learning rate and why they perform well in certain settings. Despite these optimizers being somewhat popular, we are not aware of prior work that relates them to the dynamics of standard optimizers.
> >
> > We do agree that it would be interesting to see whether longer training affect these trends and we will try to run select configurations to answer this.
> > However, we may not be able to obtain these results before the end of the discussion period.
> > We will try to do the same for the wrapped Adam+L2 on DeiT-tiny i1k.
> >
> > ## Other Questions:
> > **Figure 4:** Here each line can be thought of as a neuron that is normalized with batch normalization.
> > The results would look very similar for a whole layer when combined with layer normalization (or any "atomic" scale-invariant weight vector, i.e. those that can not be split into smaller scale-invariant vectors).
> > We will clarify this and just mention "neuron with batch normalization" in the caption, thank you.
> >
> > **Lion RMS update:** Note that the final steps in Lion are to take the elementwise sign of the update direction and then scaling this with $\eta$. Every element of the update is therefore $\pm \eta$, giving a fixed norm of $\eta \sqrt{C}$. The RMS of the (constant) update norm is therefore just equal to this value. In AdamW the size of the first moment tracker depends on $\beta_1$ despite the use of a proper exponentially weighted moving average. In a random walk, subsequent gradient vectors will add up in an orthogonal manner rather than linearly causing this effect (we discuss this more in our analysis).
> >
> > **Zero-shot RV performance on IWSLT2014 de-en:** Yes, thank you for pointing this out. We had meant to discuss this further but it somehow slipped our mind. The reason for the poor zero-shot performance is that the original network is trained with minimal weight decay (almost zero). Our analysis predicts that the rate of convergence towards equilibrium has an exponential dependency on the product of weight decay and learning rate, which is very small in this case. This prevents the network from reaching equilibrium during training, the equilibrium weight is very large and the weights only grow slowly in comparison. Consequently, the RV enforces rotational updates that significantly differ from the original training. To align the RV, we increased the weight decay value to ensure angular updates in range with those observed in the original training.

---

> > > ### Author Response · Authors · 2023-11-14
> > >
> > > ## Our Contributions
> > > We would like to emphasize what we consider the main contributions of our work and why we believe they are significant.
> > > * We derive the equilibrium states for new optimizers like AdamW. This offers valuable insights into the training dynamics of modern neural networks like transformers. The role of weight decay when optimizing these networks is not well-understood, despite its common usage (as seen in the widespread adoption of AdamW).
> > > * We empirically demonstrate that imbalanced rotation often leads to poorer performance compared to balanced rotation.
> > > Particularly, we demonstrate that AdamW results in balanced rotation unlike Adam with L2 regularization and
> > > that this rotational behavior may play a key role in the effectiveness of AdamW compared to Adam+L2.
> > > We believe this is a novel observation with potential practical and theoretical significance, warranting further investigation.
> > > * We explain how the granularity of scale-invariance matters. This can provide an explanation for the benefits of weight normalization or weight standardization in addition to e.g. layer normalization. It could also offer additional insights into why weight standardization helps in the normalization free setting (e.g. in the highly performant NF-Nets). We believe that previous research has not thoroughly explored this aspect.
> > > * We discuss how weight decay and learning rate interactions lead to a modified effective learning rate schedule, potentially contributing to the need for learning rate warmup.
> > > * We show that the effective update sizes of weights and biases have a different dependency on the weight decay and learning rate. This provides increased understanding of the roles of these hyperparameters and poses questions about how they should be varied across batch sizes or time.
> > > * We explain how weight decay also impacts the rotation of unnormalized weights, something we believe prior research has not addressed. This makes our insights applicable to standard networks that are typically not perfectly scale-invariant.
> > >
> > > **We believe each of these points is novel and relevant to the community and that as a whole our manuscript provides a new and insightful perspective on weight decay and the learning dynamics of modern deep networks.**

---

> ### Author Response · Authors · 2023-11-21
>
> We hope this message finds you well. We understand your time is precious and deeply appreciate your efforts in this review process. Since the discussion period is ending soon, we would like to kindly ask for your feedback on our rebuttal. If you feel it addresses at least some of your concerns, we would greatly appreciate it if you updated your score accordingly.
>
> Please let us know if you have any further questions and we will quickly get back to you.

---

> > ### Comment · Reviewer_tLyV · 2023-11-22
> > **Response to author rebuttal**
> >
> > Thank you for the rebuttal. I understand that the rotational analysis helps in understanding the effects of weight decay on optimizers and you do not claim to create a SOTA optimizer. I acknowledge that the procedure to create rotational variants of various existing optimizers is non-trivial. But, without studying the effects of this phenomena for larger setups, it is unclear if this holds universally. For instance, it is unclear if you run the ImageNet-1k experiments for long, do you observe the same trend? Similarly for IWSLT2014 de-en, you do not observe this effect.
> >
> > Regarding compute restrictions, you have already done ResNet50 experiments for 10 epochs, you could choose to go with much smaller architecture and show the trend with and without the rotational variants. I don't think your paper is shooting for state-of-the-art results, all you need is a trend comparing the optimizers with and without rational variants.
> >
> > Finally, the paper organization could be improved further to remove misunderstandings that a lot of reviewers had during this review/rebuttal phase.
> >
> > Given the current state, I do believe this work would benefit from another pass at it and in its current state, it has a lot of questions unanswered.

---

### Official Review · Reviewer_Yv1K · 2023-11-03

**Soundness:** 3 good
**Presentation:** 3 good
**Contribution:** 3 good
**Rating:** 5
**Confidence:** 4

**Summary:**

This paper investigates the impact of weight decay on the optimization dynamics of deep neural networks. It focuses on the concept of "equilibrium," where the effects of weight decay and gradient updates on the magnitude of parameter vectors cancel out on average, leading to stable parameter rotations and magnitudes during training. The paper explores how different optimization algorithms, including AdamW and SGD with momentum, interact with weight decay, initialization, normalization techniques, and learning rate schedules to achieve equilibrium states. The authors demonstrate that enforcing rotational equilibrium throughout training can simplify the training dynamics by eliminating chaotic transient phases. Additionally, they discuss the role of rotational behavior in the effectiveness of AdamW compared to Adam with L2-regularization, the performance of different normalization layers, and the need for learning rate warmup.

**Strengths:**

The paper studies the concept of equilibrium in the context of weight decay and optimization dynamics of SGDM and Adam. A good extension over Wan et al. 2020.

In-Depth Analysis: The paper provides a comprehensive analysis of how various factors, including optimization algorithms, weight decay, initialization, normalization, and learning rate schedules, interact and affect equilibrium states. The analysis though of over-simply assuming *random walk* updates gives estimation of the *expected angular* and *equilibrium norm*. It is interesting to see these measures are correlated with the observational phenomena of AdamW vs Adam+L2, normalization and warmup.

Moreover, the findings have practical implications for deep learning practitioners, as they suggest ways to simplify and improve training dynamics. Overall, the paper presents an intriguing concept and thorough analysis of equilibrium states in deep learning optimization.

**Weaknesses:**

**Gap between theory and practice** The expected angular exhibits different behaviors for different algorithms, e.g., Adam, AdamW and SGDM, empirically (see Figure 4 and 6). At the same time, the analysis based "random walk" gives estimation of *expected angular* and *equilibrium norm* (see Table 1). However, the theoretical estimation cannot predict the empirical differences. This makes the paper not coherent enough.

**Induced algorithm shows some improvement but not significant enought**. It is a bit disappointing that the theory motivated modification Rational Variants cannot compensate Adam+L2 to make it as good as AdamW.

**Questions:**

Can the equilibrium argument explain the practical tuning wisdom of keeping $\eta\lambda$ constant? Does this wisdom apply to all optimizers?

---

> ### Author Response · Authors · 2023-11-13
>
> Thank you for taking the time to review our paper and your feedback. We will try to address your concerns below.
>
> ## Gap between theory and practice
> **We believe there may be some misunderstanding here: we find that our theoretical predictions closely match empirical observations and can explain the differences between various optimization algorithms.** In the following, we highlight and clarify the results in our paper that support this claim, focusing on Figures 4 and 6.
>
> ### Figure 4
> This figure shows the measured weight norm and angular updates for AdamW and Adam+L2, in a simple system undergoing a random walk with a constant learning rate.
> We note that these are standard optimizers, not the rotational variants, and will therefore have a transient phase before converging to equilibrium.
> Our predicted values are derived for the steady state (equilibrium) that occurs after the initial transient phase.
> The equilibrium values also describe the expected or average behavior, the exact values fluctuate between steps although their expectation (over different random seeds or time) remains constant.
> We therefore don't expect the depicted predictions to match early in training and the exact instantaneous values in equilibrium can also deviate from them, both are fully consistent with our analysis.
>
> For **AdamW** all neurons (solid colored lines) and their average (solid black line) converge to the same weight magnitude (left) and angular update (right). In the equilibrium phase, the measurements and our predictions (dashed red line) closely align. In particular, the average (black line) aligns well with our predictions. This is because for AdamW, neurons behave similarly and the black line represents the average expected value.
>
> For **Adam+L2** every neuron converges to a different value due to the dependency on the average gradient magnitude. Although not shown in Table 1, Appendix section E thoroughly explains (predicts) this.
>
> We omitted the neuron predictions for Adam + L2 due to their complex dependency on the average gradient norm of each neuron. In a slightly simplified experimental setup, where the gradient distribution is similar across the coordinates (synapses) of each neuron, we can predict the equilibrium values from measured gradient norms.
>
> ### Figure 6
> We make the following two observations.
>
> **Left:** After extensive tuning, Adam+L2 fails to match AdamW's performance, supporting Loshchilov and Hutter's original findings in their 2019 AdamW paper.
>
> **Right:** Adam+L2 shows unbalanced average angular updates across layers, unlike AdamW. This aligns with our theory, which predicts such imbalance for Adam+L2 due to varying gradient magnitudes across layers, but not for AdamW.
> We make this observation for the optimal parameter settings for each optimizer.
>
> **General remarks:** The angular updates change over time because of the cosine learning rate schedule, the initial transient phase, and towards the end of training, as explained in detail in Section 5 (**Scheduling Effects**).

---

> > ### Author Response · Authors · 2023-11-13
> >
> > ## Performance of Rotational Variants
> > **It appears this might also be a misunderstanding: Our experiments (Table 2) show that Adam+L2 with rotational wrapper closely matches the performance of AdamW, even without additional learning rate or weight decay tuning.** This suggests that the key difference between AdamW and Adam+L2 is the balanced rotation, rather than the direction of the update, which also varies slightly.
> >
> > Our primary aim with the rotational variants (RVs) is to study the learning dynamics of standard optimizers, not introducing a new state of the art optimizer. Thus, we designed the RVs to closely mimic the equilibrium dynamics of standard optimizers, when training properly normalized networks. By design, they should therefore roughly match the performance of standard optimizers, except in cases where the original optimizers deviate significantly from the regular "balanced" equilibrium dynamics. In these cases, rotational variants tend to work better. This is evident from our experiments training poorly normalized networks and investigating the need for warm up in Section 5.
> >
> > ## Constant $\eta \lambda$
> > Yes, we believe our results show why keeping the product $\eta \lambda$ constant can lead to similar results for SGDM, AdamW, and Lion. According to Table 1, a constant $\eta \lambda$ results in the same expected angular update size $\widehat{\eta_r}$ in equilibrium. $\widehat{\eta_r}$ can be considered as a proxy for the "effective learning rate" of the weights.
> >
> > In Section 5, we discuss secondary effects that explain why configurations with a constant $\eta \lambda$ don't yield exactly the same results:
> > (1) The update size of other parameters like biases and gains differs (**Learning rate vs Weight Decay**)
> > (2) The “effective learning rate schedule” differs (**Scheduling Effects**)
> >
> > **Adam+L2:** Figure 6 (left) suggests that for Adam+L2, this does not hold. The best configurations for Adam+L2 do not lie on a proper diagonal line corresponding to a constant $\eta \lambda$, unlike for AdamW.
> >
> > ## General Remark
> > Please, let us know if you have any further questions. Aside from the concerns discussed above, your review is quite positive. If you find that our response alleviates your concerns, we would greatly appreciate it if you would consider raising your review score to reflect this.

---

> ### Author Response · Authors · 2023-11-21
>
> We hope this message finds you well. We understand your time is precious and deeply appreciate your efforts in this review process. Since the discussion period is ending soon, we would like to kindly ask for your feedback on our rebuttal. If you feel it addresses at least some of your concerns, we would greatly appreciate it if you updated your score accordingly.
>
> Please let us know if you have any further questions and we will quickly get back to you.

---

### Official Review · Reviewer_Pvr2 · 2023-11-09

**Soundness:** 3 good
**Presentation:** 3 good
**Contribution:** 2 fair
**Rating:** 5
**Confidence:** 4

**Summary:**

This submission follows the Spherical Motion Dynamics (Wan et al., 2021) to further investigate the rotational equilibrium when using normalization techniques and weight decay jointly. Some findings are further shown for analyzing the balanced and imbanced rotation, particularly for AdamW and Adam+L2.

**Strengths:**

- Simple and intuitive derivation of the "rotational equilibrum" for different optimizers
- Based on the balanced and imbanlaced rotation, interesting discovery for discribing the difference between AdamW and Adam+L2.
- Propose a simple normalization technique to explicitly control the average angular update size, forcing it to match equilibrium throughout training.

**Weaknesses:**

- I acknowledge the contribution that analyzing the difference between between AdamW and Adam+L2 from the imbalanced rotation view. However, further careful justification is required. First, the rigorous explaination is required to show why AdamW is imbanlaced while Adam+L2 is not. Second, why the balanced rotation helps to perform well in general. Without these justification, I cannot see the significant contribution of this paper, since other contribution points are too incremental and minor.

- There is a number typo in Table2, row IWSLT2014 de-en.

**Questions:**

see above

---

> ### Author Response · Authors · 2023-11-13
>
> Thank you for taking the time to review our paper and your feedback. We will try to address your concerns below.
>
> ## AdamW vs Adam+L2
> **We would like to note that we do analyze the rotational equilibrium of Adam+L2 in Appendix E. Our analysis shows that the rotation depends on the gradient magnitude, resulting in imbalanced rotation when it varies between neurons / layers.**
> This differs from AdamW, where our analysis suggests that the expected average rotation is independent of the gradient norm in equilibrium
>
> **Summary of Appendix E:**
> For Adam+L2, the equilibrium weight magnitude depends non-trivially on the average gradient norm.
> With some simplifying assumptions, we can describe the equilibrium weight magnitude as the roots of a third order polynomial.
> The closed form solution is quite long and complicated.
> We were not able to simplify it much in the general case, so we omit it from Table 1.
> The dependency on the gradient norm does not cancel out when computing the expected angular update in equilibrium.
> This differs from SGD where the equilibrium weight magnitude also depends on the average gradient norm, but this dependency cancels out when computing the average angular update.
>
> Please let us know if you find this explanation sufficient or if you have any suggestions for how it could be improved.
> We believe we could also give a more high-level intuitive explanation for the dependency on the gradient magnitude for a specific setting if that would be of interest.
>
> ## Why does balanced rotation matter?
> **We note that we do provide some high level intuitions for why balanced updates might be a good heuristic in the intro and why rotation is a natural measurement of an effective update in Section 2.**
>
> First we would like to highlight that the standard practice of using a single learning rate across all neurons / layers is a practical but somewhat arbitrary choice.
> In theory, we could tune a separate learning rate and even a lr schedule for each component (although intractable in practice).
> We know that a single learning rate is often not optimal, for example AdamW can be seen as SGDM with an adaptive (time varying) learning rate scaling for each coordinate and significantly outperforms SGDM on many problems.
>
> **Similarly to the coordinate-wise scaling in AdamW, balanced rotation can be seen as a heuristic that empirically performs well.**
> Despite its performance, we don't believe balanced rotation is likely to be theoretically optimal.
> Tuning a separate learning rate for each neuron on a given problem seems unlikely to result in perfectly balanced rotation.
>
> **We would also like to emphasize that the idea of balanced rotation comes from analyzing the learning dynamics arising from good modern deep learning practices (use of normalization, weight decay etc), it does not come out of nowhere.**
> Our research provides a detailed analysis and empirical evidence to support the effectiveness of balanced rotation.
> We observe that imbalanced rotation, as seen with methods like Adam+L2 and coarse normalization, often results in poorer performance compared to when rotations are balanced (e.g. through the use of the rotational optimizer variants).
> We also show that artificially imbalancing the rotation results in worse performance.
>
> We believe that rigorously proving why balanced rotation often outperforms imbalanced rotation is likely to be a challenging theoretical problem.
> Other forms of coordinate or component wise scaling for example in AdamW remain poorly understood theoretically despite significant effort from the community.
> **Nevertheless, we believe these are valuable observations of practical relevance even if we cannot offer a theoretical explanation for why (perfectly) balanced rotation performs well. Our observations may inspire further theoretical research to explore why and when balanced rotation helps.**

---

> > ### Author Response · Authors · 2023-11-13
> >
> > ## Other contributions
> > We respectfully disagree with the view that our paper only offers minor contributions beyond analyzing the differences between AdamW and Adam+L2 through imbalanced rotation. We believe our work significantly enhances the understanding of weight decay in deep learning.
> > * We derive the equilibrium states for new optimizers like AdamW. This offers valuable insights into the training dynamics of modern neural networks like transformers. The role of weight decay when optimizing these networks is not well-understood, despite its common usage (as seen in the widespread adoption of AdamW).
> > * We empirically demonstrate that imbalanced rotation often leads to poorer performance compared to balanced rotation. We believe this is a novel observation with potential practical and theoretical significance, warranting further investigation.
> > * We explain how the granularity of scale-invariance matters. This can provide an explanation for the benefits of weight normalization or weight standardization in addition to e.g. layer normalization. It could also offer additional insights into why weight standardization helps in the normalization free setting (e.g. in the highly performant NF-Nets). We believe that previous research has not thoroughly explored this aspect.
> > * We discuss how weight decay and learning rate interactions lead to a modified effective learning rate schedule, potentially contributing to the need for learning rate warmup.
> > * We show that the effective update sizes of weights and biases have a different dependency on the weight decay and learning rate. This provides increased understanding of the roles of these hyperparameters and poses questions about how they should be varied across batch sizes or time.
> > * We explain how weight decay also impacts the rotation of unnormalized weights, something we believe prior research has not addressed. This makes our insights applicable to standard networks that are typically not perfectly scale-invariant.
> >
> > **We believe each of these points is novel and relevant to the community and that as a whole our manuscript provides a new and insightful perspective on weight decay and the learning dynamics of modern deep networks.**
> >
> > ## Typo in Table 2
> > Thank you for pointing out the zero-shot performance of the RV. This is not a typo, but rather an oversight in our discussion.
> > The reason for the poor zero-shot performance is that the original network is trained with minimal weight decay (almost zero).
> > Our analysis predicts that the rate of convergence towards equilibrium has an exponential dependency on the product of weight decay and learning rate, which is very small in this case.
> > This prevents the network from reaching equilibrium during training, the equilibrium weight is very large and the weights only grow slowly in comparison.
> > Consequently, the RV enforces rotational updates that significantly differ from the original training.
> > To align the RV, we increased the weight decay value to ensure angular updates in range with those observed in the original training.

---

> ### Author Response · Authors · 2023-11-21
>
> We hope this message finds you well. We understand your time is precious and deeply appreciate your efforts in this review process. Since the discussion period is ending soon, we would like to kindly ask for your feedback on our rebuttal. If you feel it addresses at least some of your concerns, we would greatly appreciate it if you updated your score accordingly.
>
> Please let us know if you have any further questions and we will quickly get back to you.

---

### Public Comment · ~Zeke_Xie1 · 2023-11-19
**Related Work on Dynamics of Weight Decay**

Dear Authors,

I read your ICLR submission with great interest.

It is a great contribution to understanding weight decay and how it affects deep learning dynamics.

I would like to draw your attention to one of my recent works which I think is very relevant.

[1], NeurIPS2023, studied the overlooked pitfalls of weight decay (especially for Adam) from a perspective of gradient norms.
A number of interested observations (about BatchNorm/convergence/generalization/Minima Sharpness) were reported.


Reference:

[1] Xie, Z., Zhang, J., Sato, I., & Sugiyama, M. (2023, November). On the Overlooked Pitfalls of Weight Decay and How to Mitigate Them: A Gradient-Norm Perspective. In Thirty-seventh Conference on Neural Information Processing Systems.

---

> ### Author Response · Authors · 2023-11-19
>
> Thank you, we find your work very interesting as well and will make sure to discuss/cite it in a future version of our manuscript.

---

### Author Response · Authors · 2023-11-20
**Updated Revision**

We have uploaded a revised version of our manuscript. Aside from minor fixes, there are two main changes:
1) We found a new approximation for the Adam+L2 that gives us interpretable closed form expressions for the equilibrium weight norm and angular updates (instead of the third order polynomials we had before). We have added this to Table 1. We hope this makes it clearer that our analysis can explain why Adam+L2 results in imbalanced rotation unlike AdamW.
2) We have updated our explanation of the random walk setting we use in our analysis to better relate it to real neural network training (modifying the beginning of Section 3).

---

> ### Author Response · Authors · 2023-11-21
>
> Please note that we have uploaded a new, revised version of our manuscript. In this version, we have **highlighted the changes from the last update in light red** for easier identification.

---

### Meta-Review · Area_Chair_QxWo · 2023-12-10

**Metareview:**

This paper characterizes the "Rotational Equilibria" in optimization algorithms when weight decay is used. It considers commonly used optimizers including AdamW and SGD with momentum. The paper shows that enforcing rotational equilibrium throughout training can simplify the dynamics by eliminating transient phases. The paper also compares AdamW to Adam with L2 regularization from this perspective.

The reviewers appreciated the novel contributions of this paper, especially the part comparing AdamW to Adam with L2 regularization. However, they all had concerns about the paper and leaned towards rejection. In particular, one of the common concerns is what are the advantages of the proposed RVs. The reviewers felt that the provided evidence was insufficient to convey its significance. Additionally, the paper structure and presentation clarity could be improved as the reviewers appeared to have a lot of misunderstanding.

**Justification For Why Not Higher Score:**

See meta-review.

**Justification For Why Not Lower Score:**

N/A

---

### Decision · Program_Chairs · 2024-01-16

Reject